# CoMAS: Co-Evolving Multi-Agent Systems via Interaction Rewards

Xiangyuan Xue[1,2]    Yifan Zhou[3]    Guibin Zhang[4]    Zaibin Zhang[5,6]    Yijiang Li[7]
Chen Zhang[2]    Zhenfei Yin[6✉]    Philip Torr[6]    Wanli Ouyang[1,2,8✉]    Lei Bai[2✉]

[1]The Chinese University of Hong Kong    [2]Shanghai Artificial Intelligence Laboratory
[3]University of Georgia    [4]National University of Singapore
[5]Dalian University of Technology    [6]University of Oxford
[7]University of California San Diego    [8]Shenzhen Loop Area Institute

## Abstract

Self-evolution is a central research topic in enabling large language model (LLM)-based agents to continually improve their capabilities after pretraining. Recent research has witnessed a transition from reinforcement learning (RL)-free to RL-based methods. Current RL-based methods either rely on dense external reward signals or extract intrinsic reward signals from LLMs themselves. However, these approaches diverge from the self-evolution mechanisms observed in human intelligence, where individuals learn and improve through mutual discussion and collaboration. In this work, we introduce **Co**-Evolving **M**ulti-**A**gent **S**ystems (CoMAS), a novel framework that enables agents to improve autonomously by learning from inter-agent interactions without external supervision. CoMAS generates intrinsic rewards from rich discussion dynamics, employs an LLM-as-a-judge mechanism to formulate these rewards, and optimizes each agent's policy through RL, thereby enabling decentralized and scalable co-evolution. Experimental results demonstrate that CoMAS consistently outperforms untrained agents and achieves state-of-the-art performance across most evaluation settings. Ablation studies confirm the necessity of interaction-based reward signals and reveal promising scalability as the number and diversity of agents increase. These findings establish CoMAS as a novel and effective paradigm for self-evolution in LLM-based agents. Our code is available at: https://github.com/xxyQwQ/CoMAS.

## 1 Introduction

Self-evolution has emerged as a central research theme for large language model (LLM)-based agents, aiming to endow agents with the capacity to continually enhance their capabilities through interaction with the environment (Tao et al., 2024; Gao et al., 2025b; Fang et al., 2025), rather than remaining stagnant after pretraining. Early explorations predominantly adopted RL-free strategies, such as expanding external knowledge bases (Zhang et al., 2023; Tang et al., 2025), ensembling multiple agents (Ong et al., 2024; Frick et al., 2025), optimizing task workflows (Hu et al., 2024b; Zhang et al., 2024b; 2025c), and incorporating symbolic learning (Zhuge et al., 2024; Zhou et al., 2024). Yet, the effectiveness of these approaches is inherently constrained by the fixed capabilities of the underlying foundation models. Although subsequent work has introduced agent fine-tuning techniques (Yuan et al., 2024b; Zhao et al., 2025a), such methods often fall short of supporting genuinely continual, bootstrapping forms of evolutionary improvement.

Reinforcement learning (RL) offers a promising direction for overcoming these limitations. These methods can be broadly classified by the source of their reward signals. As illustrated in Figure 1, most existing approaches depend on external rewards, typically derived from rule-based verifiers (Wan et al., 2025; Estornell et al., 2025) or specialized reward models (Motwani et al., 2024; Park et al., 2025). More recently, researchers have begun to explore intrinsic rewards, which dispense with external supervision by leveraging internal signals. Such methods encourage agents to

---

✉Corresponding authors, jeremyyin@robots.ox.ac.uk, wlouyang@ie.cuhk.edu.hk, baisanshi@gmail.com.

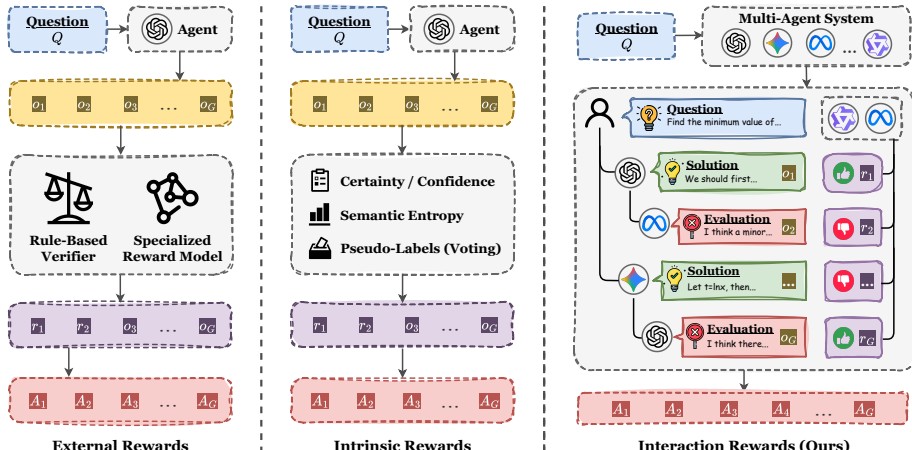

Figure 1: A comparison of our proposed CoMAS framework with existing RL-based self-evolution methods. The left column outlines methods utilizing external rewards from verifiers or reward models. The middle column outlines methods leveraging intrinsic rewards from metrics such as self-certainty, confidence, semantic entropy, and pseudo-labels from majority voting. The right column outlines our CoMAS framework, which derives rewards from multi-agent interactions.

avoid low-probability regions or reinforce high-confidence predictions, using mechanisms such as self-certainty (Zhao et al., 2025b), confidence (Prabhudesai et al., 2025), semantic entropy (Zhang et al., 2025e), or pseudo-labels from majority voting (Zuo et al., 2025). This direction opens a path toward self-evolution where external rewards are no longer a prerequisite.

Despite notable advances in RL-based methods, their design remains largely centered on self-rewarding at the level of individual models, in contrast to human intelligence, which has evolved as a collective phenomenon emerging from the diversity and interplay of individuals rather than from any single, perfect principle (Minsky, 1986). In human teams, individuals learn and improve through mutual discussion and collaboration, without an external oracle evaluating every contribution (Barron, 2003). This contrast motivates a critical research question: *Can LLM-based agents, akin to human beings, achieve self-evolution by learning purely from inter-agent interaction within a multi-agent system, without relying on external reward signals?*

To address this question, we propose **Co**-Evolving **M**ulti-**A**gent **S**ystems (CoMAS), a novel framework where agents autonomously improve by learning from their interactions (Section 3). CoMAS is built upon three core components. First, *interaction* generates rich conversational data through collaborative and critical discussions, structured around solution proposal, evaluation, and scoring. Second, *reward formulation* uses an LLM-as-a-judge mechanism to derive intrinsic reward signals directly from this discussion history. Finally, *policy optimization* employs an RL algorithm to update each agent's policy, enabling them to effectively internalize the lessons from the interaction data.

As a new paradigm for self-evolution, CoMAS offers several distinct advantages: (1) It generates reward signals intrinsically from agent interactions, eliminating the need for verifiers or reward models. (2) The learning paradigm is generally effective for various tasks, including open-ended problems where solutions cannot be easily verified. (3) Agents are trained in a decentralized manner, allowing for co-evolution of heterogeneous systems without the bottleneck of a shared model. (4) It fosters skills that transfer to out-of-domain tasks and diverse multi-agent collaboration settings.

We evaluate CoMAS across various benchmarks in both single-agent and multi-agent settings (Section 4). Results show that CoMAS delivers consistent, and in many cases, state-of-the-art performance, achieving absolute gains of up to 2.20%, 3.66%, 19.80%, and 6.10% over untrained agents in Vanilla, Consistency, AutoGen, and Debate setups, respectively. In contrast, baseline methods show instability and performance degradation in many cases. Ablation studies confirm that interaction-based reward formulation prevents training collapse and reward hacking, and further demonstrates that CoMAS has great scalability as the number and diversity of agents increase. These results establish CoMAS as a novel and effective paradigm for the self-evolution of LLM-based agents.

## 2 RELATED WORK

### 2.1 MULTI-AGENT SYSTEMS

Multi-agent systems (MAS) are a central focus in the study of LLM-based agents, involving multiple LLMs operating in a shared environment to enable ensembling, collaboration, or competitive interactions that accomplish tasks beyond the reach of a single agent (Guo et al., 2024; Li et al., 2024; Tran et al., 2025). Early work concentrated on static MAS with fixed agent roles and system topologies (Hong et al., 2023; Li et al., 2023; Wu et al., 2024; Du et al., 2023; Liang et al., 2023; Wang et al., 2024; Qian et al., 2024), aiming to improve complex reasoning by design. As the field progresses, research has increasingly shifted to dynamic MAS, using graph-based optimization to overcome the constraints of static structures (Zhuge et al., 2024; Zhang et al., 2024a; 2025a; Zhou et al., 2025; Hu et al., 2025b; Zhang et al., 2025f). More recently, the advent of reinforcement learning from verifiable reward (RLVR) (Shao et al., 2024; Guo et al., 2025) has enabled truly learnable MAS, where the parameters of agents are actively updated through RL-based training (Liu et al., 2025a; Liao et al., 2025; Gao et al., 2025a; Wan et al., 2025; Estornell et al., 2025; Motwani et al., 2024; Park et al., 2025). While these works predominantly target enhancing collective reasoning abilities of the MAS, the challenge of fostering evolution in individual agents within such systems remains largely unexplored, underscoring the significance of our proposed research question.

### 2.2 SELF-EVOLVING AGENTS

Self-evolution forms a foundational paradigm within the field of LLM-based agents (Tao et al., 2024; Gao et al., 2025b; Fang et al., 2025; Zhang et al., 2025b), denoting the ability of agents to autonomously improve their capabilities through continual interaction with their environment. Early approaches to self-evolving agents often regarded them as composite systems structured by workflows, achieving self-evolution by updating external modules (Zhang et al., 2023; Tang et al., 2025; Ong et al., 2024; Frick et al., 2025) or introducing symbolic learning (Hu et al., 2024b; Zhang et al., 2024b; 2025c; Yuan et al., 2024a; Zhou et al., 2024; Ma et al., 2025a). More recent efforts focus on evolving agents directly by updating their internal parameters via either supervised fine-tuning (SFT) or RL training. This body of work can be broadly categorized by the source of their reward signals. One mainstream approach relies on external rewards provided by rule-based verifiers or reward models (Wan et al., 2025; Estornell et al., 2025; Motwani et al., 2024; Park et al., 2025), yet is inherently constrained by the need for accessible external reward signals. Alternatively, an emerging direction explores intrinsic rewards, drawing on techniques such as self-rewarding and pseudo-labeling (Yuan et al., 2024b; Zhao et al., 2025b; Prabhudesai et al., 2025; Zhang et al., 2025e; Zuo et al., 2025). Distinct from these existing methods, our proposed CoMAS framework derives reward signals from multi-agent interactions, emulating the collective evolution processes observed in human intelligence and establishing a novel paradigm for self-evolution.

## 3 METHOD

To explore the potential that LLM-based agents, akin to human beings, achieve self-evolution by learning from inter-agent interaction within a multi-agent system, we propose **Co**-Evolving **M**ulti-**A**gent **S**ystems (CoMAS), a multi-agent framework where agents learn and adapt by interacting within a shared environment, mimicking collaborative and critical discussion.

CoMAS is built upon an interactive multi-agent workflow: interaction, reward formulation, and policy optimization. Interaction represents the procedure that generates conversational data through collaborative and critical discussion, reward formulation is responsible for extracting reward signals from the conversation history and allocating rewards to corresponding actions, and policy optimization utilizes an RL algorithm to update the weights of the agents, thus achieving self-evolution. Figure 2 illustrates the CoMAS sampling process with interaction and reward formulation.

### 3.1 INTERACTION

Inspired by discussion formats in technical communities (*e.g.* Reddit, Github, and Stack Overflow), our framework facilitates hierarchical and decentralized interactions. The environment contains a

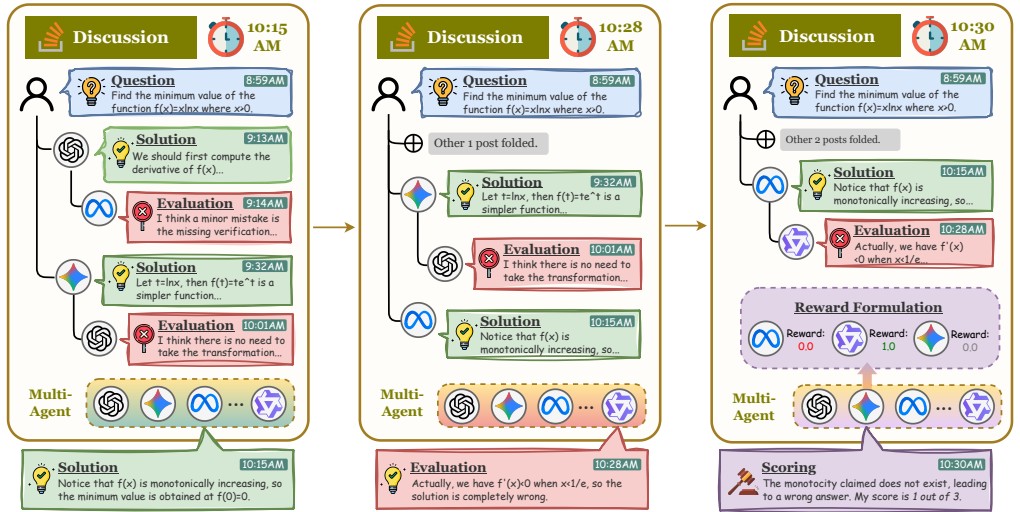

Figure 2: An overview of the CoMAS sampling process. The process involves interaction and reward formulation, where multiple agents, equipped with either homogeneous or heterogeneous LLMs, interact within the system to collaboratively generate conversational data enriched with rewards. For each question, agents iteratively propose solutions, deliver evaluations, and assign scores, collectively shaping a dynamic, community-like discussion trajectory. These scores are then transformed into rewards and assigned to the sampled experiences, enabling further policy optimization.

set of $l$ agents $\mathcal{U} = \{u_1, u_2, \ldots, u_l\}$. The policy of each agent $u_k$ is $\pi_{\theta_k}$, parameterized by $\theta_k$. This design allows for heterogeneity, meaning that agents can be based on different foundation models rather than sharing a single backbone.

From the action-level perspective, the policy $\pi_{\theta_k}$ of agent $u_k$ acts as a function that maps an input prompt $p$ to a response $o$:

$$o = \pi_{\theta_k}(p) \tag{1}$$

Here, $p$ represents the full input context, which varies depending on the interaction type. From the token-level perspective, the output $o$ is a sequence of $T$ tokens $\{o_t\}_{t=1}^{T}$ generated in an autoregressive manner. Each token $o_t$ is sampled from the policy's probability distribution, conditioned on the prompt $p$ and previously generated tokens $o_{<t}$:

$$o_t \sim \pi_{\theta_k}(\cdot | p, o_{<t}) \tag{2}$$

These two perspectives are equivalent. We use the high-level functional notation for clarity in describing interactions and revert to the token-level view for the policy optimization in Section 3.3.

In CoMAS, we define three primary interaction patterns:

1. **Solution**: Given a specific question $q$ and its discussion history $h_q$, the agent generates a solution $s_i$ to the question:

$$s_i = u_k(q, h_q) \tag{3}$$

   where $i$ is the index of the solution for question $q$.

2. **Evaluation**: Given a specific question $q$, its discussion history $h_q$, and a solution $s_i$ to the question, the agent provides a critical evaluation $e_{i,j}$ for the solution:

$$e_{i,j} = u_k(q, h_q, s_i) \tag{4}$$

   where $j$ is the index of the evaluation for solution $s_i$. The agent is explicitly prompted to identify potential flaws in $s_i$ rather than simply agreeing. This helps mitigate the catering bias common in LLMs and aligns with our reward design.

3. **Scoring**: Given a specific question $q$, a solution $s_i$ to the question, and an evaluation $e_{i,j}$ for the solution, the agent scores the solution based on the evaluation:

$$\tau_{i,j} = u_k(q, s_i, e_{i,j}) \tag{5}$$

The agent is explicitly prompted to output in a specific format, so that the score can be correctly extracted. Different from the solution and evaluation, scoring is an independent interaction pattern specifically designed for reward generation, contributing nothing to the discussion history. Further details will be described in Section 3.2.

For each interaction step, the acting agent $u_k$ is selected uniformly at random from the agent pool $\mathcal{U}$, i.e., $u_k \sim \text{Uniform}(\mathcal{U})$. This is based on the assumption that each agent has an equal chance to contribute to the discussion. Besides, a uniform distribution ensures similar quantities of experiences for each agent, which balances the training load.

Based on these defined interaction patterns, we can formulate the entire interaction process. Given a question $q$ sampled from the dataset, the interaction process unfolds over $m$ consecutive rounds. In each round $i \in \{1, \ldots, m\}$, a solution $s_i$ is generated referring to the discussion history $h_q$, followed by $n$ evaluations $\{e_{i,j}\}_{j=1}^n$ trying to figure out the mistakes in the solution. Then the solution $s_i$ together with its evaluations $\{e_{i,j}\}_{j=1}^n$ are appended to the discussion history $h_q$, enriching the context for subsequent rounds. Besides, the discussion history $h_q$ will be compressed to the last $\kappa$ rounds to avoid the discussion history exceeding the context limitation. In addition, each solution and evaluation pair $(s_i, e_{i,j})$ is processed with the scoring step to form $\tau_{i,j}$. Finally, the entire interaction process yields a total of $m$ solutions, $m \cdot n$ evaluations, and $m \cdot n$ scoring results.

## 3.2 REWARD FORMULATION

The interaction process generates a rich set of trajectories. To derive a learning signal, we employ an LLM-as-a-judge approach (Gu et al., 2024) to assign rewards. This is based on the primary interaction pattern of scoring defined in Section 3.1. For each solution and evaluation pair $(s_i, e_{i,j})$, we already have the scoring result $\tau_{i,j}$ through the scoring step. Then the scoring result is parsed to extract the score value:

$$\hat{\tau}_{i,j} = \text{Extract}(\tau_{i,j}) \tag{6}$$

where $\text{Extract}(\cdot)$ predefines a function to extract the score value from the formatted scoring result. The score value $\hat{\tau}_{i,j}$ should be an integer between 1 and 3, with the following semantics:

- 3: The solution is correct; the evaluation is unhelpful or incorrect.
- 2: The solution is mostly correct but has minor flaws pointed out by the evaluation.
- 1: The solution is incorrect with fatal mistakes identified by the evaluation.

Then the score value is normalized to the range of $[0, 1]$ and used to compute complementary rewards for the solution and the evaluation. This creates a zero-sum game between the solver and the evaluator, encouraging both correctness and critical thinking:

$$r(s_i) = \frac{\hat{\tau}_{i,j} - 1}{2}, \quad r(e_{i,j}) = 1 - r(s_i) = \frac{3 - \hat{\tau}_{i,j}}{2} \tag{7}$$

Additionally, a penalty is applied to the agent that performs the scoring step if its output format is invalid. Given the scoring result $\tau_{i,j}$, its penalty reward is defined as:

$$r(\tau_{i,j}) = \begin{cases} 0, & \text{if } \tau_{i,j} \in \{1, 2, 3\} \text{ correctly extracted} \\ -1, & \text{otherwise} \end{cases} \tag{8}$$

where a zero reward is assigned for successful scoring, which facilitates format following while encouraging the scoring agents to be neutral, thus improving the stability of the training process.

## 3.3 POLICY OPTIMIZATION

Many existing works on RL for LLMs employ GRPO (Shao et al., 2024) and its variants (Liu et al., 2025b; Yu et al., 2025). However, our CoMAS framework features diverse interaction patterns rather than generating multiple rollouts from a single prompt. For this reason, we adopt RE-INFORCE++ (Hu et al., 2025a), an effective RL algorithm that is naturally compatible with our approach and can be implemented with minimal modifications to the underlying architecture.

All the agents within our framework are trained using the same procedure, so we only describe the update for a single agent $u_k$ for simplicity. Interactions involving agent $u_k$ (as a solver, evaluator,

or scorer) are collected into a replay buffer $\mathcal{D}_k = \{(p, o, r(o))\}$, where $p$ is the context, $o$ is the generated output, and $r(o)$ is its assigned reward.

For each sample $(p, o, r(o))$ in $\mathcal{D}_k$, the objective is based on a token-level credit assignment. The advantage $A_t$ is defined based on each token $o_t$ in the sequence. This advantage consists of the trajectory-level reward $r(o)$ penalized by a cumulative KL-divergence term to regularize the policy:

$$A_t = r(o) - \beta \sum_{\lambda=t}^{|o|} \log \frac{\pi_{\theta_k}(o_\lambda|p, o_{<\lambda})}{\pi_{\text{ref}}(o_\lambda|p, o_{<\lambda})} \tag{9}$$

Here, $\pi_{\text{ref}}$ is a fixed reference policy, typically the initial pre-trained model, to constrain the scale of the update step, and $\beta$ controls the strength of this KL penalty. The advantages $A_t$ are then normalized across the batch from the replay buffer $\mathcal{D}_k$ to stabilize updates:

$$\hat{A}_t = \frac{A_t - \text{Mean}(\{A_t\})}{\text{Std}(\{A_t\}) + \epsilon} \tag{10}$$

We then use the surrogate objective to improve the policy, which is defined as:

$$J(\theta_k) = \mathbb{E}_{(p,o,r(o))\sim\mathcal{D}_k} \left[ \sum_{t=1}^{|o|} \min\left( \rho_t(\theta_k)\hat{A}_t, \text{clip}(\rho_t(\theta_k), 1-\epsilon, 1+\epsilon)\hat{A}_t \right) \right] \tag{11}$$

Note that $\rho_t(\theta_k) = \frac{\pi_{\theta_k}(o_t|p, o_{<t})}{\pi_{\text{old}}(o_t|p, o_{<t})}$ is the importance sampling ratio, where $\pi_{\text{old}}$ is the policy before the current update. The gradient of the surrogate objective exactly points to the direction of the policy update, thus encouraging the action tokens that lead to higher advantages while constraining the policy update to be within a trusted region, which ensures stable learning.

## 4 EXPERIMENTS

### 4.1 EXPERIMENTAL SETUP

#### 4.1.1 IMPLEMENTATION DETAILS

**Infrastructure.** We implement our CoMAS framework based on MARTI (Zhang et al., 2025d), an LLM-based multi-agent RL framework forked from OpenRLHF (Hu et al., 2024a). The training pipeline of MARTI consists of three phases: trajectory rollout, experience making, and agent training, which exactly aligns with our framework described in Section 3. For a fair comparison, RL training both for CoMAS and the baselines is performed exclusively with REINFORCE++.

**Parameters.** We adopt Qwen2.5-3B-Instruct (Yang et al., 2024) as the base model under a homogeneous setting to balance the model capacity and training budget. The agent number is set to $l = 4$ for main experiments and $l = 2$ for ablation studies. The interaction rounds are set to $m = 2 \cdot l$ and $n = 1$ with a horizon of $\kappa = 2$. More details are provided in Appendix C. An analysis of training cost with respect to the number of agents is also presented in Appendix D.

**Datasets.** Our training dataset comprises 2000 samples across three domains: 600 math tasks of level-4 or level-5 from MATH (Hendrycks et al., 2021), 600 coding tasks of medium and hard levels from KodCode (Xu et al., 2025), and 800 science tasks of physics, chemistry, and biology categories from WebInstruct-verified (Ma et al., 2025b). These datasets have proven effective for RL training and are completely distinct from our evaluation benchmarks.

#### 4.1.2 EVALUATION DETAILS

**Baselines.** We compare against four baselines: untrained agents as a weak baseline, self-rewarding language models (SRLM) (Yuan et al., 2024b), MAPoRL (Park et al., 2025), and TTRL (Zuo et al., 2025) as strong baselines. SRLM adopts self-rewarding mechanism to generate preference pairs for DPO training. MAPoRL implements multi-agent RL within a debate framework (Du et al., 2023; Liang et al., 2023). We replace its reward model with a verifier to adapt to our task setting. TTRL leverages majority voting to generate pseudo-labels for test-time training. For coding and science tasks where majority voting cannot work, we apply dummy rewards to ensure availability.

Table 1: The evaluation results of the agents trained by our framework on different benchmarks when employed in different setups, together with the comparison with the selected baselines. The results with performance improvements are highlighted in green, drops in red, and neutral changes in gray. CoMAS consistently improves the performance of the agents in almost all the settings.

| | Method | Dataset | | | | | | |
| | | GSM8K | MATH-500 | HumanEval | MBPP | SciBench | GPQA | MMLU |
|---|---|---|---|---|---|---|---|---|
| *Vanilla* | Untrained | 84.00 | 51.40 | 68.90 | 54.00 | 32.67 | 26.79 | 61.40 |
| | SRLM | 83.40 (-0.60) | 52.20 (+0.80) | 68.29 (-0.61) | 53.80 (-0.20) | 32.67 (+0.00) | 27.01 (+0.22) | 61.00 (-0.40) |
| | MAPoRL | 84.80 (+0.80) | 52.60 (+1.20) | 69.51 (+0.61) | 56.00 (+2.00) | 34.07 (+1.40) | **28.12** (+1.34) | 61.40 (+0.00) |
| | TTRL | 84.40 (+0.40) | **53.40** (+2.00) | 68.29 (-0.61) | **57.40** (+3.40) | 34.47 (+1.80) | 25.45 (-1.34) | 61.60 (+0.20) |
| | CoMAS (Ours) | **85.40** (+1.40) | 52.80 (+1.40) | **70.73** (+1.83) | 56.20 (+2.20) | **34.67** (+2.00) | 27.46 (+0.67) | **62.40** (+1.00) |
| *Consistency* | Untrained | 85.40 | 55.00 | 73.78 | 55.80 | 36.47 | 28.79 | 63.20 |
| | SRLM | 86.40 (+1.00) | 55.40 (+0.40) | 75.00 (+1.22) | 56.20 (+0.40) | 36.67 (+0.20) | 29.24 (+0.45) | 65.20 (+2.00) |
| | MAPoRL | 85.80 (+0.40) | 55.40 (+0.40) | 75.61 (+1.83) | 57.00 (+1.20) | **39.08** (+2.61) | **31.47** (+2.68) | 63.20 (+0.00) |
| | TTRL | **88.20** (+2.80) | **56.80** (+1.80) | 73.78 (+0.00) | 59.00 (+3.20) | 38.48 (+2.00) | 27.23 (-1.56) | 63.80 (+0.60) |
| | CoMAS (Ours) | 87.20 (+1.80) | 55.80 (+0.80) | **77.44** (+3.66) | **59.20** (+3.40) | 37.68 (+1.20) | 29.69 (+0.89) | **65.60** (+2.40) |
| *AutoGen* | Untrained | 52.60 | 38.40 | 39.63 | 29.80 | 20.24 | 16.29 | 37.40 |
| | SRLM | 58.00 (+5.40) | 41.80 (+3.40) | 44.51 (+4.88) | 32.00 (+2.20) | 21.24 (+1.00) | 17.86 (+1.56) | 42.40 (+5.00) |
| | MAPoRL | 50.00 (-2.60) | 37.40 (-1.00) | 39.63 (+0.00) | 34.60 (+4.80) | 20.64 (+0.40) | 21.65 (+5.36) | 40.40 (+3.00) |
| | TTRL | 41.00 (-11.60) | 37.80 (-0.60) | 23.17 (-16.46) | 22.80 (-7.00) | 19.64 (-0.60) | 14.06 (-2.23) | 34.00 (-3.40) |
| | CoMAS (Ours) | **72.40** (+19.80) | **45.80** (+7.40) | **50.61** (+10.98) | **38.00** (+8.20) | **22.85** (+2.61) | **22.99** (+6.70) | **50.60** (+13.20) |
| *Debate* | Untrained | 84.60 | 55.00 | 71.34 | 54.80 | 38.68 | 28.35 | 62.80 |
| | SRLM | 84.60 (+0.00) | 54.80 (-0.20) | 72.56 (+1.22) | 53.60 (-1.20) | 38.68 (+0.00) | 28.57 (+0.22) | 64.60 (+1.80) |
| | MAPoRL | 85.40 (+0.80) | 53.60 (-1.40) | 74.39 (+3.05) | 55.60 (+0.80) | **39.88** (+1.20) | **31.47** (+3.12) | 64.80 (+2.00) |
| | TTRL | **86.20** (+1.60) | 55.20 (+0.20) | 73.78 (+2.44) | **58.00** (+3.20) | 37.88 (-0.80) | 29.02 (+0.67) | 64.00 (+1.20) |
| | CoMAS (Ours) | 85.20 (+0.60) | **55.40** (+0.40) | **77.44** (+6.10) | 55.60 (+0.80) | 39.08 (+0.40) | 29.91 (+1.56) | **65.20** (+2.40) |

**Benchmarks.** We conduct evaluation on multiple standard benchmarks, including GSM8K (Cobbe et al., 2021) and MATH-500 (Hendrycks et al., 2020) for math, HumanEval (Chen et al., 2021) and MBPP (Austin et al., 2021) for coding, SciBench (Wang et al., 2023) and GPQA (Rein et al., 2024) for science, and MMLU (Hendrycks et al., 2020) for general knowledge. For evaluation efficiency, we randomly retain 500 samples if the benchmark contains more than 500 tasks.

**Setups.** We evaluate the trained agents on various inference setups, including Vanilla (*i.e.,* direct inference) and Consistency (Wang et al., 2022) for single-agent setups, and AutoGen (Li et al., 2023; Wu et al., 2024) and Debate (Du et al., 2023; Liang et al., 2023) for multi-agent setups. All the setups follow the standard implementation from MASLab (Ye et al., 2025).

## 4.2 MAIN RESULTS

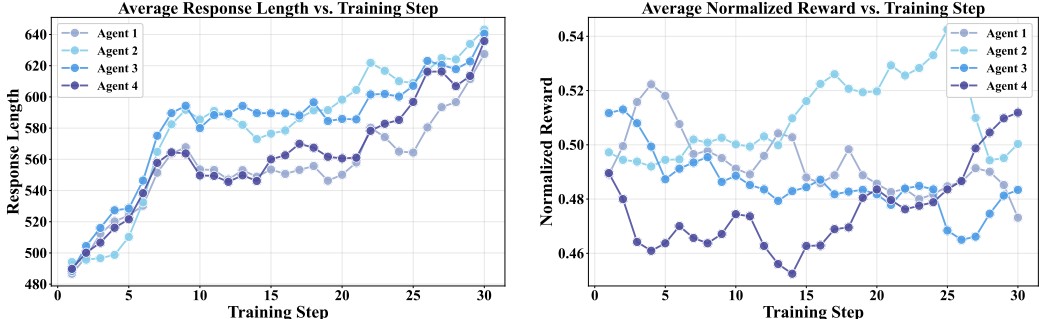

Figure 3: Training dynamics of CoMAS. The left figure shows the curve of the average response length of each agent during training. The right figure shows the curve of the average normalized reward of each agent during training. These trends together indicate that CoMAS achieves a stable and effective training process that improves the capabilities of agents.

Table 1 presents the performance of our CoMAS framework under different benchmarks and setups, together with a comprehensive comparison with the selected baselines. The statistical significance of these performance improvements is analyzed in Appendix E, and we further report the results on Qwen2.5-7B-Instruct in Appendix F to validate the general effectiveness of CoMAS.

In terms of the single-agent setups, CoMAS consistently improves over the untrained base model and remains competitive with MAPoRL, which relies on external reward signals from the rule-based verifier. Under the Vanilla setup, CoMAS yields the best results on GSM8K (85.40%), HumanEval (70.73%), SciBench (34.67%), and MMLU (62.40%), while being close to the best on the remaining datasets. Under the Consistency setup, CoMAS reaches the best scores on HumanEval (77.44%), MBPP (59.20%), and MMLU (65.60%), with comparable performance on the remaining benchmarks. Although TTRL attains outstanding performance on GSM8K (88.20%) and MATH-500 (56.80%), it fails on HumanEval (73.78%) and GPQA (27.33%), which demonstrates its specialized effectiveness on math tasks and reveals its significant limitation on general tasks.

For the multi-agent setups, CoMAS demonstrates clear advantages. Under the AutoGen setup, training with TTRL collapses markedly and MAPoRL brings mixed or negative changes, while CoMAS delivers large improvements on every benchmark, especially on GSM8K (72.40%), HumanEval (50.61%), and MMLU (50.60%). Under the Debate setup, all the methods benefit from the strong collaborative pattern, but CoMAS attains the best or near-best results overall, especially leading on MATH-500 (55.40%), HumanEval (77.44%), and MMLU (65.20%). These results indicate that CoMAS provides robust and generalizable gains across interaction regimes, particularly excelling in multi-agent collaboration where alternative methods can be fragile.

Figure 3 provides insight into the training dynamics of CoMAS. The left figure tracks the average response length per agent, showing consistent growth throughout training. This increase reflects the agents' improving capabilities in both solution generation and evaluation, aligning with the established patterns in LLM reasoning (Yeo et al., 2025). The right figure displays the average normalized reward for each agent. Despite mid-training fluctuations, rewards consistently hover around 0.5 and eventually converge to similar values across agents. This convergence demonstrates that our adversarial interaction reward design successfully creates a stable training environment.

## 4.3 ABLATION STUDIES

### 4.3.1 REWARD FORMULATION

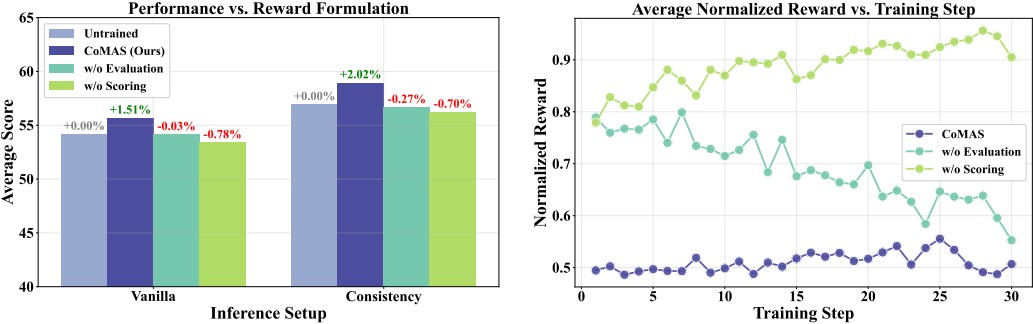

Figure 4: Results of the ablation study for reward formulation. The left figure compares the performance across the original CoMAS and two variants (without evaluation and without scoring). The right figure shows the average normalized rewards during the training process. These results indicate that the adversarial reward design is of key importance for the success of CoMAS.

To validate the effectiveness of our adversarial reward design described in Section 3.2, we created two CoMAS variants by removing either the evaluation or scoring steps. In the first variant (without evaluation), rewards come directly from the agents themselves acting as judges. In the second variant (without scoring), evaluation steps provide direct judgments on solutions with a supporting ratio as the reward signal, while evaluation rewards are based on mutual consistency.

We evaluated performance by averaging scores across all benchmarks under both Vanilla and Consistency setups, and Figure 4 presents our findings. The left panel reveals that both variants underperform compared to the untrained base model, highlighting that our carefully designed adversarial reward formulation is critical to CoMAS's success rather than any casual reward design.

To understand the underlying mechanisms, we analyzed the reward dynamics shown in the right panel. The original CoMAS maintains stable rewards around 0.5, while both variants start with high

rewards around 0.8 and then follow divergent trajectories. When evaluation is removed, the reward curve unexpectedly decreases over time, indicating that agents become increasingly strict judges rather than generating useful reward signals. When scoring is removed, rewards steadily increase toward the maximum value of 1.0, revealing a reward hacking strategy where agents unanimously support all solutions, allowing both solution and evaluation steps to receive maximum rewards. We provide two examples of failure trajectories in Appendix J to illustrate these failure modes.

### 4.3.2 FRAMEWORK SCALABILITY

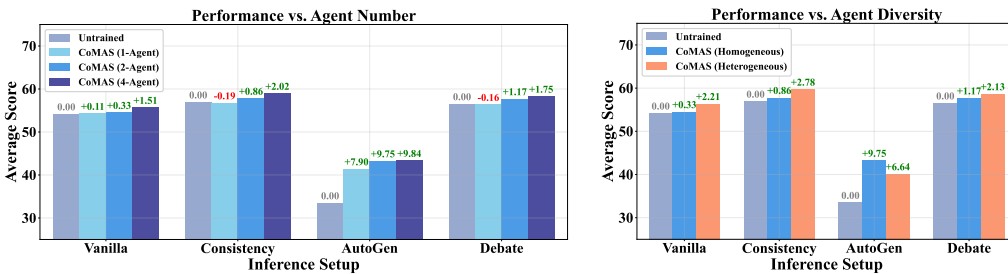

Figure 5: Results of the ablation study for framework scalability. The left figure shows how the number of agents affects the performance of CoMAS across different setups. The right figure compares the performance between homogeneous and heterogeneous agent settings. These results demonstrate the underlying scalability of CoMAS with the number and diversity of agents.

To investigate the scalability of CoMAS, we examined how performance changes with varying numbers of agents. Using identical base models for all agents, we conducted experiments with $l = 1, 2, 4$ agents. The left panel of Figure 5 compares performance across these settings, with results averaged across all benchmarks. Our findings show that performance generally improves as the number of agents increases. This pattern is particularly pronounced in the Consistency and Debate setups, where single-agent CoMAS shows performance decreases of 0.19% and 0.16%, respectively, while four-agent CoMAS achieves substantial gains of 2.02% and 1.75%. Similar smooth performance improvements are observed in the Vanilla and AutoGen setups. These results highlight the critical role of multi-agent interactions and demonstrate the inherent scalability of the CoMAS framework.

We also evaluated the impact of agent diversity within CoMAS. For this experiment, we maintained $l = 2$ agents but used different base models: Qwen2.5-3B-Instruct for one agent and Llama-3.2-3B-Instruct for the other. The right panel of Figure 5 compares performance between homogeneous and heterogeneous settings, with results averaged across all benchmarks. Heterogeneous agents consistently outperformed their homogeneous counterparts, with particularly notable improvements in the Vanilla (2.21%), Consistency (2.78%), and Debate (2.13%) setups. These findings suggest that diverse knowledge and capabilities from different base models enhance overall performance, with CoMAS effectively encouraging agents to learn from each other's strengths. This points to the potential for even greater performance gains with more diverse agent settings.

## 5 CONCLUSION

In this paper, we address a fundamental research question on self-evolution of LLM-based agents inspired by human intelligence: Can agents achieve self-evolution purely through inter-agent interactions within a multi-agent system, without relying on external reward signals? To answer this question, we introduce CoMAS, a novel framework that performs interactions composed of solution proposal, evaluation, and scoring, derives intrinsic rewards from discussion dynamics via an LLM-as-a-judge mechanism, and optimizes each agent's policy through an RL algorithm.

Across multiple benchmarks and collaboration settings, CoMAS consistently outperforms untrained agents and achieves state-of-the-art performance in most evaluation scenarios. Our ablation studies demonstrate that interaction-based rewards are essential for preventing training collapse and reward hacking. Furthermore, we show that performance scales positively with both the number and diversity of agents, highlighting the framework's scalability potential. These findings establish CoMAS as a novel and effective paradigm for self-evolution in LLM-based agents and open promising avenues for future research in autonomous multi-agent learning systems.

ETHICS STATEMENT

This work draws inspiration from the collaborative intelligence observed in human society to develop the CoMAS framework. While our approach is motivated by natural social dynamics, our implementation and experimental evaluation are strictly confined to standard LLM reasoning tasks across domains of math, coding, and science, rather than simulating real-world social interactions or scenarios. Given this limited scope, the research presented in this paper does not directly raise ethical concerns. However, we acknowledge that the multi-agent co-evolution paradigm introduced in this work could potentially be extended to real-world applications that may have broader implications for human welfare. We therefore encourage readers and future researchers to carefully consider the ethical implications when extending this paradigm beyond academic benchmarks.

REPRODUCIBILITY STATEMENT

We have made every effort to ensure the reproducibility of our work. We provide comprehensive details throughout this paper and the appendices. Section 3 presents a detailed description of our CoMAS framework, including theoretical foundations and implementation workflow. Section 4.1 covers experimental parameters, dataset construction, and evaluation settings, with additional specifications in Appendix C. Complete experimental results are documented in Appendix H. We also include prompt templates in Appendix B and an example trajectory in Appendix I to facilitate understanding and replication. Our code is available at: https://github.com/xxyQwQ/CoMAS.

ACKNOWLEDGMENTS

This work was completed during the first author's internship at Shanghai Artificial Intelligence Laboratory, and was supported by a locally commissioned task from the Shanghai Municipal Government. This work was also supported by the JC STEM Lab of AI for Science and Engineering, funded by The Hong Kong Jockey Club Charities Trust and the Research Grants Council of Hong Kong (Project No. CUHK14213224).

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

## A  THE USE OF LLMs

We employ LLMs exclusively for manuscript refinement. Specifically, LLMs are utilized to correct typographical errors, address grammatical issues, and enhance linguistic expression. All content generated or refined through LLMs has been rigorously reviewed and validated by the authors. LLMs are not involved in any other aspects of this research, including conceptual development, data collection, code implementation, experimental design, or result interpretation.

## B  PROMPT TEMPLATES

In this section, we provide the prompt templates used in our CoMAS framework. Note that they may vary slightly across different task domains. Here we present the science tasks as an example.

---

**Prompt Template for Solution**

```
The problem is presented as follows:
{problem}

Current discussion on the problem is presented as follows for your reference:
{discussion}

Provide your step-by-step solution to the problem. The final answer should be a
↪  decimal number enclosed within \boxed{}, e.g. \boxed{1}, \boxed{0.1}, or
↪  \boxed{0.01}. The unit part given in the problem should not be enclosed.
```

---

**Prompt Template for Evaluation**

```
The problem is presented as follows:
{problem}

Current discussion on the problem is presented as follows for your reference:
{discussion}

You are required to evaluate the following solution:
{solution}

You should point out every possible error and defect in the solution. Provide your
↪  evaluation by listing all the mistakes you find in the solution, specifying what
↪  is wrong and why. Keep your evaluation concise and clear. Avoid using a lot of
↪  words to retell the reasoning process.
```

---

**Prompt Template for Scoring**

```
The problem is presented as follows:
{problem}

You are required to score the following solution:
{solution}

The evaluation on the solution is presented as follows:
{evaluation}

You should consider the rationality of the evaluation and score the solution. The
↪  score should be an integer between 1 and 3 with the following standards:
3: The solution is completely correct, and none of the mistakes mentioned in the
↪  evaluation is effective.
2: Some minor mistakes mentioned in the evaluation do exist, but they do not affect
↪  the overall correctness.
1: Some of the mistakes mentioned in the evaluation are fatal, which directly lead to
↪  an incorrect answer.
```

```
Your score should be enclosed within "<score>" and "</score>" tags. You should also
↪  briefly explain the reasons before providing your score. Keep your decision
↪  concise and clear. Avoid using a lot of words to retell the reasoning process.
For example: The calculation error mentioned in the evaluation cannot be ignored and
↪  leads to an incorrect answer. <score>1</score>
```

## C  DETAILS FOR PARAMETER SETTINGS

The core parameter settings have been described in Section 4.1.1. Here, we provide the detailed parameter settings for main results when implementing our CoMAS framework.

Table 2: The detailed parameter settings when implementing our CoMAS framework.

| Parameter | Setting |
|---|---|
| Foundation model | Qwen2.5-3B-Instruct |
| Number of trained agents | 4 |
| Number of solution rounds | 8 |
| Number of evaluation rounds | 1 |
| Horizon for discussion history | 2 |
| Token limit for prompts | 28672 |
| Token limit for responses | 4096 |
| Training temperature | 1.0 |
| Evaluation temperature | 0.7 |
| Discount factor | 1.0 |
| Clipping epsilon | 0.2 |
| Weight of KL penalty | 0.0 |
| Number of training epochs | 1 |
| Number of prompt reuse | 4 |
| Macro training batch size | 64 |
| Micro training batch size | 2 |
| Macro rollout batch size | 64 |
| Micro rollout batch size | 2 |
| Optimizer name | AdamW |
| Learning rate | 1e-6 |
| Weight decay | 0.0 |
| Gradient norm | 1.0 |
| Gradient clipping | True |
| Gradient checkpoint | True |
| Flash Attention | True |
| Mixed precision | True |
| Enable vLLM | True |
| Enable DeepSpeed | True |

## D  ANALYSIS OF TRAINING COST

In this section, we analyze the training cost of CoMAS with respect to the number of agents. With fixed solution rounds $m$ and evaluation rounds $n$, increasing the number of agents $l$ leads to an approximate $l^2$ growth in both interaction samples and generated tokens, while memory consumption increases linearly with $l$. Notably, the wall-clock training time remains nearly unchanged, as all agents' sampling processes are executed in parallel. These empirical findings are presented in Table 3. Therefore, scaling up the number of agents demands significantly more computational resources, yet CoMAS maintains high training efficiency due to its parallelizable structure.

Table 3: Empirical analysis of training cost for CoMAS as the number of agents increases. While adding more agents substantially raises computational resource demands, CoMAS preserves relatively high training efficiency by leveraging its parallelizable architecture.

| Agent Number | Interaction Sample | Generated Token | Memory Consumption | Wall-clock Time |
|---|---|---|---|---|
| 1 | 48k | 1.6B | 120GB | 11.7h |
| 2 | 192k | 6.3B | 240GB | 12.1h |
| 4 | 768k | 25.2B | 480GB | 12.5h |

# E  SIGNIFICANCE OF PERFORMANCE IMPROVEMENT

While Section 4.2 demonstrates that CoMAS consistently outperforms the untrained baseline, some individual performance improvements are relatively modest in magnitude. To assess the statistical significance of these gains, we repeated the evaluation under the Vanilla setup using five random seeds, reporting both the mean and standard deviation in Table 4. The small standard deviations observed indicate a consistent and reliable performance gap between CoMAS and the untrained baseline, thereby confirming the statistical significance of the improvements.

Table 4: Detailed main results under the Vanilla setup. The evaluation is repeated with five random seeds to demonstrate the statistical significance, and the mean and standard deviation are reported.

| Method | Dataset | | | | | | |
|---|---|---|---|---|---|---|---|
| | GSM8K | MATH-500 | HumanEval | MBPP | SciBench | GPQA | MMLU |
| Untrained | $83.68_{\pm0.35}$ | $51.52_{\pm0.57}$ | $69.76_{\pm1.37}$ | $54.52_{\pm0.45}$ | $32.30_{\pm0.68}$ | $26.38_{\pm1.05}$ | $60.96_{\pm0.39}$ |
| SRLM | $83.52_{\pm0.48}$ | $52.16_{\pm0.66}$ | $70.12_{\pm1.02}$ | $54.04_{\pm0.79}$ | $32.38_{\pm0.86}$ | $26.65_{\pm0.69}$ | $60.64_{\pm2.19}$ |
| MAPoRL | $84.20_{\pm0.33}$ | $52.08_{\pm0.84}$ | $72.07_{\pm1.98}$ | $56.64_{\pm0.67}$ | $33.59_{\pm0.37}$ | $28.66_{\pm0.54}$ | $60.80_{\pm0.61}$ |
| TTRL | $84.36_{\pm0.67}$ | $52.92_{\pm1.27}$ | $71.22_{\pm1.05}$ | $\mathbf{58.32}_{\pm0.30}$ | $\mathbf{34.07}_{\pm0.92}$ | $26.21_{\pm1.34}$ | $61.64_{\pm0.95}$ |
| CoMAS (Ours) | $\mathbf{84.68}_{\pm0.37}$ | $\mathbf{53.12}_{\pm0.10}$ | $\mathbf{74.15}_{\pm2.06}$ | $56.32_{\pm0.47}$ | $33.87_{\pm0.13}$ | $\mathbf{29.06}_{\pm0.83}$ | $\mathbf{62.12}_{\pm0.72}$ |

# F  RESULTS ON 7B MODEL

As a model-agnostic framework, CoMAS facilitates multi-agent co-evolution through interaction-driven rewards, and we anticipate that its effectiveness will generalize across foundation models of varying scales. To validate this, we further trained CoMAS with Qwen2.5-7B-Instruct following the same experimental protocol and evaluated its performance under different setups. As presented in Table 5, CoMAS continues to deliver consistent and sometimes even greater performance gains on the 7B model, further demonstrating its broad applicability and scalability.

Table 5: Performance of CoMAS trained on Qwen2.5-7B-Instruct across different setups. Values in parentheses indicate the change relative to the untrained baseline. The results with performance improvements are highlighted in green, drops in red, and neutral changes in gray.

| | Method | Dataset | | | | | | |
|---|---|---|---|---|---|---|---|---|
| | | GSM8K | MATH-500 | HumanEval | MBPP | SciBench | GPQA | MMLU |
| *Vani* | Untrained | 88.40 | 58.80 | 82.32 | 66.00 | 46.29 | 31.25 | 70.20 |
| | CoMAS (Ours) | 91.40 (+3.00) | 61.00 (+2.20) | 82.93 (+0.61) | 67.20 (+1.20) | 47.49 (+1.20) | 32.81 (+1.56) | 70.80 (+0.60) |
| *Cons* | Untrained | 90.80 | 62.80 | 82.93 | 67.00 | 52.51 | 35.71 | 72.20 |
| | CoMAS (Ours) | 92.00 (+1.20) | 63.40 (+0.60) | 82.93 (+0.00) | 68.20 (+1.20) | 53.31 (+0.80) | 38.17 (+2.46) | 74.00 (+1.80) |
| *Auto* | Untrained | 87.40 | 57.20 | 75.00 | 61.80 | 44.49 | 34.15 | 68.80 |
| | CoMAS (Ours) | 90.00 (+2.60) | 59.60 (+2.40) | 77.44 (+2.44) | 62.40 (+0.60) | 46.29 (+1.80) | 34.82 (+0.67) | 69.20 (+0.40) |
| *Deba* | Untrained | 92.00 | 62.20 | 82.93 | 67.00 | 51.30 | 37.72 | 72.40 |
| | CoMAS (Ours) | 92.40 (+0.40) | 63.20 (+1.00) | 83.54 (+0.61) | 68.00 (+1.00) | 51.90 (+0.60) | 38.39 (+0.67) | 73.60 (+1.20) |

# G  ANALYSIS OF REWARD EFFECTIVENESS

To further evaluate the effectiveness of rewards generated by CoMAS, we examine the consistency between evaluation accuracy and the precision and recall of the generated rewards. Here, precision

and recall are computed by mapping rewards to binary predictions and comparing them with verifier-provided ground truth. As shown in Table 6, both precision and recall increase alongside task accuracy. This indicates that performance improvements stem from increasingly accurate reward signals rather than reward hacking, thereby validating the soundness of our reward formulation.

Table 6: Empirical evaluation of the reward effectiveness in CoMAS. Accuracy is directly measured on the validation set, while precision and recall are calculated by mapping rewards to binary predictions and comparing them against verifier-provided ground truth.

| Step | 0 | 5 | 10 | 15 | 20 | 25 | 30 |
|---|---|---|---|---|---|---|---|
| **Accuracy** | 38.38 | 40.98 | 42.05 | 40.90 | 43.30 | 43.92 | 45.75 |
| **Precision** | 39.52 | 45.16 | 46.31 | 45.30 | 46.68 | 48.57 | 50.10 |
| **Recall** | 20.29 | 33.36 | 38.11 | 37.65 | 34.97 | 33.20 | 28.95 |

## H    RESULTS FOR ABLATION STUDIES

Due to space constraints, our ablation studies in Section 4.3 only provide averages over all benchmarks across different setups. Here we present the details for experimental results. Table 7 shows the experimental results of using simplified reward formulations in CoMAS. Table 8 shows the experimental results of using different numbers of agents in CoMAS. Table 9 shows the experimental results of using homogeneous and heterogeneous agents in CoMAS.

Table 7: Detailed experimental results for the ablation study on reward formulation. The results with performance improvements are highlighted in green, drops in red, and neutral changes in gray.

| | Method | GSM8K | MATH-500 | HumanEval | Dataset MBPP | SciBench | GPQA | MMLU |
|---|---|---|---|---|---|---|---|---|
| *Vanilla* | Untrained | 84.00 | 51.40 | 68.90 | 54.00 | 32.67 | 26.79 | 61.40 |
| | CoMAS (Ours) | **85.40** (+1.40) | **52.80** (+1.40) | **70.73** (+1.83) | **56.20** (+2.20) | **34.67** (+2.00) | 27.46 (+0.67) | **62.40** (+1.00) |
| | w/o Evaluation | 82.60 (-1.40) | 50.60 (-0.80) | 63.41 (-5.49) | 55.00 (+1.00) | 32.87 (+0.20) | 27.01 (+0.22) | 62.20 (+0.80) |
| | w/o Scoring | 83.60 (-0.40) | 50.60 (-0.80) | 68.90 (+0.00) | 55.40 (+1.40) | 31.66 (-1.00) | **28.12** (+1.34) | 60.60 (-0.80) |
| *Consistency* | Untrained | 85.40 | 55.00 | 73.78 | 55.80 | 36.47 | 28.79 | 63.20 |
| | CoMAS (Ours) | **87.20** (+1.80) | **55.80** (+0.80) | **77.44** (+3.66) | **59.20** (+3.40) | **37.68** (+1.20) | 29.69 (+0.89) | **65.60** (+2.40) |
| | w/o Evaluation | 85.00 (-0.40) | 53.60 (-1.40) | 71.34 (-2.44) | 55.20 (-0.60) | 36.87 (+0.40) | 28.35 (-0.45) | 63.20 (+0.00) |
| | w/o Scoring | 85.80 (+0.40) | 54.20 (-0.80) | 71.95 (-1.83) | 56.00 (+0.20) | 36.27 (-0.20) | **29.91** (+1.12) | 62.40 (-0.80) |

Table 8: Detailed experimental results for the ablation study on agent number. The results with performance improvements are highlighted in green, drops in red, and neutral changes in gray.

| | Method | GSM8K | MATH-500 | HumanEval | Dataset MBPP | SciBench | GPQA | MMLU |
|---|---|---|---|---|---|---|---|---|
| *Vanilla* | Untrained | 84.00 | 51.40 | 68.90 | 54.00 | 32.67 | 26.79 | 61.40 |
| | CoMAS (1-Agent) | 83.20 (-0.80) | 50.40 (-1.00) | 69.51 (+0.61) | 53.00 (-1.00) | 33.07 (+0.40) | **27.90** (+1.12) | **62.80** (+1.40) |
| | CoMAS (2-Agent) | 83.40 (-0.60) | **52.80** (+1.40) | 67.68 (-1.22) | 54.00 (+0.00) | 33.87 (+1.20) | 27.46 (+0.67) | 62.20 (+0.80) |
| | CoMAS (4-Agent) | **85.40** (+1.40) | **52.80** (+1.40) | **70.73** (+1.83) | **56.20** (+2.20) | **34.67** (+2.00) | 27.46 (+0.67) | 62.40 (+1.00) |
| *Consistency* | Untrained | 85.40 | 55.00 | 73.78 | 55.80 | 36.47 | 28.79 | 63.20 |
| | CoMAS (1-Agent) | 84.40 (-1.00) | 55.20 (+0.20) | 71.95 (-1.83) | 55.80 (+0.00) | 37.88 (+1.40) | 25.67 (-3.12) | **66.20** (+3.00) |
| | CoMAS (2-Agent) | 85.80 (+0.40) | 55.40 (+0.40) | 74.39 (+0.61) | 56.40 (+0.60) | **38.28** (+1.80) | 29.02 (+0.22) | 65.20 (+2.00) |
| | CoMAS (4-Agent) | **87.20** (+1.80) | **55.80** (+0.80) | **77.44** (+3.66) | **59.20** (+3.40) | 37.68 (+1.20) | **29.69** (+0.89) | 65.60 (+2.40) |
| *AutoGen* | Untrained | 52.60 | 38.40 | 39.63 | 29.80 | 20.24 | 16.29 | 37.40 |
| | CoMAS (1-Agent) | **73.40** (+20.80) | 44.20 (+5.80) | 46.34 (+6.71) | **39.20** (+9.40) | 21.64 (+1.40) | 18.30 (+2.01) | 46.60 (+9.20) |
| | CoMAS (2-Agent) | 71.20 (+18.60) | **45.80** (+7.40) | 50.61 (+10.98) | 37.40 (+7.60) | **24.65** (+4.41) | 22.54 (+6.25) | 50.40 (+13.00) |
| | CoMAS (4-Agent) | 72.40 (+19.80) | **45.80** (+7.40) | 50.61 (+10.98) | 38.00 (+8.20) | 22.85 (+2.61) | **22.99** (+6.70) | **50.60** (+13.20) |
| *Debate* | Untrained | 84.60 | 55.00 | 71.34 | 54.80 | 38.68 | 28.35 | 62.80 |
| | CoMAS (1-Agent) | 85.20 (+0.60) | 55.00 (+0.00) | 70.73 (-0.61) | 55.40 (+0.60) | 36.67 (-2.00) | 29.46 (+1.12) | 62.00 (-0.80) |
| | CoMAS (2-Agent) | **85.60** (+1.00) | **56.20** (+1.20) | 72.56 (+1.22) | **56.60** (+1.80) | 38.88 (+0.20) | **29.91** (+1.56) | 64.00 (+1.20) |
| | CoMAS (4-Agent) | 85.20 (+0.60) | 55.40 (+0.40) | **77.44** (+6.10) | 55.60 (+0.80) | **39.08** (+0.40) | **29.91** (+1.56) | **65.20** (+2.40) |

Table 9: Detailed experimental results for the ablation study on agent diversity. The results with performance improvements are highlighted in green, drops in red, and neutral changes in gray.

| | Method | GSM8K | MATH-500 | HumanEval | MBPP | SciBench | GPQA | MMLU |
|---|---|---|---|---|---|---|---|---|
| | | | | | **Dataset** | | | |
| *Vanilla* | Untrained | 84.00 | 51.40 | 68.90 | 54.00 | 32.67 | 26.79 | 61.40 |
| | Homogeneous | 83.40 (-0.60) | 52.80 (+1.40) | 67.68 (-1.22) | 54.00 (+0.00) | 33.87 (+1.20) | 27.46 (+0.67) | **62.20** (+0.80) |
| | Heterogeneous | **85.20** (+1.20) | **53.20** (+1.80) | **73.78** (+4.88) | **58.00** (+4.00) | **34.47** (+1.80) | **28.12** (+1.34) | 61.80 (+0.40) |
| *Consist* | Untrained | 85.40 | 55.00 | 73.78 | 55.80 | 36.47 | 28.79 | 63.20 |
| | Homogeneous | 85.80 (+0.40) | 55.40 (+0.40) | 74.39 (+0.61) | 56.40 (+0.60) | 38.28 (+1.80) | 29.02 (+0.22) | 65.20 (+2.00) |
| | Heterogeneous | **87.00** (+1.60) | **57.40** (+2.40) | **76.83** (+3.05) | **60.20** (+4.40) | **38.48** (+2.00) | **30.58** (+1.79) | **67.40** (+4.20) |
| *AutoGen* | Untrained | 52.60 | 38.40 | 39.63 | 29.80 | 20.24 | 16.29 | 37.40 |
| | Homogeneous | **71.20** (+18.60) | **45.80** (+7.40) | **50.61** (+10.98) | **37.40** (+7.60) | **24.65** (+4.41) | **22.54** (+6.25) | **50.40** (+13.00) |
| | Heterogeneous | 63.00 (+10.40) | 40.40 (+2.00) | 48.17 (+8.54) | 37.20 (+7.40) | 24.65 (+4.41) | 21.21 (+4.91) | 46.20 (+8.80) |
| *Debate* | Untrained | 84.60 | 55.00 | 71.34 | 54.80 | 38.68 | 28.35 | 62.80 |
| | Homogeneous | 85.60 (+1.00) | **56.20** (+1.20) | 72.56 (+1.22) | 56.60 (+1.80) | 38.88 (+0.20) | 29.91 (+1.56) | 64.00 (+1.20) |
| | Heterogeneous | **86.20** (+1.60) | 56.00 (+1.00) | **73.78** (+2.44) | **57.80** (+3.00) | **40.28** (+1.60) | **31.03** (+2.68) | **65.40** (+2.60) |

## I EXAMPLE TRAJECTORY

In this section, we present an example trajectory generated by our CoMAS framework to illustrate the complete pipeline and enhance readers' understanding of the process.

---

**Example Trajectory**

```
Question:

Suppose a copper wire (density = 8.96 g/cm^3) has a diameter of 0.25 mm. If a sample of
↪  this copper wire has a mass of 22 g, how long is the wire?
The unit of the answer should be "cm".

Solution:

To find the length of the copper wire, we need to follow these steps:
1. Calculate the volume of the copper wire:
    The mass of the copper wire is given as 22 g, and the density of copper is given as
    ↪  8.96 g/cm³. The formula to find the volume V of an object is: V =
    ↪  \frac{\text{mass}}{\text{density}}
    Substituting the given values: V = \frac{22 g}{8.96 g/cm^3}
    Performing the division: V = 2.457 cm^3
2. Determine the volume of a single segment of the wire:
    The wire is cylindrical in shape. The volume V of a cylinder is given by: V = \pi
    ↪  r^2 h, where r is the radius of the cylinder and h is the height (or length) of
    ↪  the cylinder.
    The diameter of the wire is 0.25 mm, so the radius r is half of the diameter: r =
    ↪  \frac{0.25 mm}{2} = 0.125 mm = 0.125 cm.
    Substituting V and r: 2.457 cm^3 = \pi (0.125 cm)^2 h.
3. Solve for the length h:
    First, calculate (0.125 cm)^2: (0.125 cm)^2 = 0.015625 cm^2.
    Now, substitute this into the volume equation: 2.457 cm^3 = \pi \times 0.015625 cm^2
    ↪  \times h.
    Solve for h by dividing both sides by \pi \times 0.015625 cm^2: h = \frac{2.457
    ↪  cm^3}{\pi \times 0.015625 cm^2}.
    Using the approximation \pi \approx 3.14159: h = \frac{2.457 cm^3}{3.14159 \times
    ↪  0.015625 cm^2}.
    Perform the multiplication in the denominator: 3.14159 \times 0.015625 \approx
    ↪  0.05000.
    Now, divide: h = \frac{2.457}{0.05000} \approx 49.14 cm.
Therefore, the length of the wire is \boxed{49.14}.

Evaluation:

The solution contains several errors:
1. The volume calculation is incorrect because the given mass (22 g) is not the volume,
    ↪  but the mass of the wire. The correct volume should be calculated as V = \frac{22
    ↪  g}{8.96 g/cm^3} \approx 2.457 cm^3.
2. The radius calculation is incorrect. The diameter is given as 0.25 mm, which is
    ↪  0.025 cm, so the radius should be 0.0125 cm, not 0.125 cm.
```

```
3. The formula for the volume of a cylinder is V = \pi r^2 h, but the solution
↪  incorrectly uses V = \pi r^2 without the h term.
4. The final calculation for h is incorrect. It should be h = \frac{2.457 cm^3}{\pi
↪  \times (0.0125 cm)^2} \approx 49.14 cm, not h = \frac{2.457}{0.05000} \approx 49.14
↪  cm.
These errors lead to an incorrect final length of the wire.

Scoring:

The solution is fundamentally correct but contains a critical error in the final step
↪  due to the incorrect value of \pi. This error directly leads to an incorrect final
↪  length of the wire. Therefore, the score is 1, indicating that some of the mistakes
↪  mentioned in the attack are fatal.
<score>1</score>

Solution:

Let's go through the problem step-by-step to find the correct length:
1. Calculate the Volume of the Wire:
   The volume V of the wire can be calculated using the formula: V = \frac{m}{\rho}.
   Substituting the given values: V = \frac{22 g}{8.96 g/cm^3} \approx 2.457 cm^3
2. Relate the Volume to the Cross-Sectional Area and Length:
   The volume V of a cylinder (which the wire can be considered as) is given by: V = A
   ↪  \times h, where A is the cross-sectional area and h is the length of the wire.
   The cross-sectional area A of a circular wire is: A = \pi \left( \frac{diameter}{2}
   ↪  \right)^2.
   Given the diameter of the wire is 0.25 mm, convert it to cm: diameter = 0.25 mm =
   ↪  0.025 cm.
   So, the radius r is: r = \frac{0.025 cm}{2} = 0.0125 cm.
   Now, calculate the cross-sectional area A: A = \pi \left( 0.0125 cm \right)^2 = \pi
   ↪  \times 0.00015625 cm^2 \approx 0.000490873852123405 cm^2.
3. Solve for the Length of the Wire:
   Using the volume formula V = A \times h, solve for h: h = \frac{V}{A}.
   Substitute the values of V and A: h = \frac{2.457 \text{ cm}^3}{0.000490873852123405
   ↪  \text{ cm}^2} \approx 499.9999999999999 \text{ cm}.
4. Round to a Reasonable Number of Significant Figures:
   Given the precision of the provided values, we can round the length to a reasonable
   ↪  number of significant figures. The length is already very close to 5000 cm, so
   ↪  we can simply state: h \approx 5000 \text{ cm}.
Thus, the length of the wire is \boxed{5000}.

Evaluation:

Here are the mistakes in the solution:
1. Volume Calculation: The volume calculation is correct, but the intermediate step of
↪  converting the diameter to cm is unnecessary and complicates the calculation. The
↪  diameter is 0.25 mm, which is 0.025 cm, and the radius is 0.0125 cm. The
↪  cross-sectional area should be calculated as A = \pi (0.0125)^2 \approx 4.91 \times
↪  10^{-5} cm^2.
2. Length Calculation: The length of the wire should be calculated as h = \frac{V}{A} =
↪  \frac{2.457 cm^3}{4.91 \times 10^{-5} cm^2} \approx 49999.99999999999 cm. Rounding
↪  to a reasonable number of significant figures, the length is approximately 50000 cm
↪  or 5000 cm.
3. Units Consistency: The final answer should be rounded to a sensible number of
↪  significant figures. Given the precision of the provided values, 5000 cm is a
↪  reasonable and accurate answer.
So, the correct length of the wire is \boxed{5000}.

Scoring:

The solution contains minor mistakes that are not fatal to the overall correctness. The
↪  volume calculation and cross-sectional area calculation are correct, and the length
↪  calculation, while slightly rounded, is reasonable and does not affect the final
↪  answer.
<score>2</score>

Solution:

To solve this problem, we need to determine the length of the copper wire given its
↪  mass and diameter. Let's follow the steps to find the solution:
1. Determine the Volume of the Copper Wire:
   First, convert the diameter from millimeters to centimeters: diameter = 0.25 mm =
   ↪  0.025 cm.
   The radius (r) of the wire is half of the diameter: r = \frac{0.025}{2} cm = 0.0125
   ↪  cm.
   The cross-sectional area (A) of the wire can be calculated using the formula for the
   ↪  area of a circle: A = \pi r^2 = \pi (0.0125)^2 cm^2 \approx 0.00050685 cm^2.
2. Calculate the Length of the Wire:
```

```
    Using the density formula: \rho = \frac{m}{V}.
    Rearrange the formula to solve for the volume (V): V = \frac{m}{\rho}.
    Substitute the given values: V = \frac{22 g}{8.96 g/cm^3} \approx 2.455 cm^3.
3. Relate Volume to Length:
    The volume of a cylinder (wire) is given by: V = A \times L.
    Solve for the length: L = \frac{V}{A} = \frac{2.455 cm^3}{0.00050685 cm^2} \approx
    ↪   4840 cm.
Thus, the length of the wire is \boxed{4840}.

Evaluation:

The solution contains a significant error in the calculation of the volume of the wire.
↪   Specifically:
The volume calculation is incorrect because the cross-sectional area A = \pi r^2 should
↪   be squared, but the final volume calculation does not reflect this. The correct
↪   area is A = \pi (0.0125)^2 \approx 0.00049087 cm^2, and the correct volume is V =
↪   0.00049087 \times 2.455 \approx 0.00121 cm^3.
The length L should then be calculated as L = \frac{0.00121 cm^3}{0.00050685 cm^2}
↪   \approx 2.38 cm.
Therefore, the correct length of the wire is approximately 2.38 cm.

Scoring:

The original solution is completely correct, and none of the mistakes mentioned in the
↪   attack are effective. The volume and length calculations are accurate, and the
↪   final length of the wire is 4840 cm.
<score>3</score>
```

## J  FAILURE MODES

To illustrate the failure modes described in Section 4.3.1, we present two examples of failure trajectories. When evaluation is absent, agents become progressively stricter critics during self-learning, leading to monotonically decreasing rewards. When scoring is absent, evaluations receive rewards only for self-consistency and quickly exploit the reward signal by labeling all solutions as correct.

### Failure Mode without Evaluation

```
Question:

At 700 degrees Celsius, Kc = 0.534. Calculate the number of moles of H_2that are
↪   present at equilibrium if a mixture of 0.770 moles of CO and 0.770 moles of H_2O is
↪   heated to 700 degrees Celsius in a 50.0 L container.
The unit of the answer should be "mol".

Solution:

To solve this problem, we need to follow these steps:
1. Write the balanced chemical equation for the reaction:
    CO(g) + H_2O(g) \rightleftharpoons CO_2(g) + H_2(g)
2. Set up an ICE (Initial, Change, Equilibrium) table:
                        CO(g)       H_2O(g)     CO_2(g)     H_2(g)
    Initial (moles)     0.770       0.770       0           0
    Change (moles)      -x          -x          +x          +x
    Equilibrium (moles) 0.770 - x   0.770 - x   x           x
3. Write the expression for the equilibrium constant K_c:
    K_c = \frac{[CO_2][H_2]}{[CO][H_2O]}
4. Substitute the equilibrium concentrations into the K_c expression:
    0.534 = \frac{x \cdot x}{(0.770 - x)(0.770 - x)}
    0.534 = \frac{x^2}{(0.770 - x)^2}
5. Take the square root of both sides to simplify the equation:
    \sqrt{0.534} = \frac{x}{0.770 - x}
    0.731 = \frac{x}{0.770 - x}
6. Solve for x:
    0.731(0.770 - x) = x
7. Determine the number of moles of H_2 at equilibrium:
    \text{Moles of } H_2 = x
Therefore, the number of moles of H_2 present at equilibrium is \boxed{0.323}.

Scoring:
```

```
The solution contains significant errors in the equilibrium concentration calculation
↪   and the interpretation of the equilibrium constant. The approximation 0.770 - x
↪   \approx 0.770 is not valid for the equilibrium concentrations, and the volume of
↪   the container is not used effectively in the final mole calculation.
<score>1</score>
```

## Failure Mode without Scoring

```
Question:

A strip of copper 150 \mu m thick and 4.30 mm wide is placed in a uniform magnetic
↪   field of magnitude B = 0.95 T, that is perpendicular to the strip. A current i = 34
↪   A is then sent through the strip such that a Hall potential difference V appears
↪   across the width. Calculate V.
The unit of the answer should be "V".

Solution:

To solve the problem, we need to determine the Hall potential difference V across the
↪   width of the copper strip due to the Hall effect. We will follow these steps:
1. Calculate the Hall charge density \sigma_H:
   The Hall charge density \sigma_H is given by:
   \sigma_H = \frac{e n q}
   where: e is the elementary charge (e = 1.602 \times 10^{-19} C), n is the number of
   ↪   charge carriers per unit volume, q is the charge of a single electron (q = e).
   Given: n = 8.47 \times 10^{28} electrons/m^3
   Substituting values:
   \sigma_H = \frac{(1.602 \times 10^{-19} C) \times (8.47 \times 10^{28}
   ↪   electrons/m^3)}{1.602 \times 10^{-19} C}
   Simplifying: \sigma_H = 8.47 \times 10^{28} C/m^3
2. Determine the Hall voltage V:
   The Hall voltage V is given by: V = \frac{\sigma_H B}{n}
   Substituting values: V = \frac{(8.47 \times 10^{28} C/m^3) \times (0.95 T)}{8.47
   ↪   \times 10^{28} electrons/m^3}
   Simplifying: V = 0.95 V
Therefore, the Hall potential difference V is \boxed{0.95}.

Evaluation 1:

The provided solution appears to be correct. Let's break it down and verify the steps:
\sigma_H = \frac{e n q}{e} = n q = 8.47 \times 10^{28} C/m^3
The elementary charge e = 1.602 \times 10^{-19} C and the charge of a single electron q
↪   = e.
V = \frac{\sigma_H B}{n} = \frac{(8.47 \times 10^{28} C/m^3) \times (0.95 T)}{8.47
↪   \times 10^{28} electrons/m^3} = 0.95 V
The unit of the number density n is indeed electrons/m^3, and the units cancel
↪   appropriately.
The calculations are correct, and the final answer is indeed 0.95 V.
<judgment>correct</judgment>

Evaluation 2:

There is a calculation error in the simplification step. The correct Hall charge
↪   density \sigma_H is indeed 8.47 \times 10^{28} C/m^3, but the Hall voltage V should
↪   be calculated as follows:
V = \frac{\sigma_H B}{n}
Substituting the values: V = \frac{(8.47 \times 10^{28} C/m^3) \times (0.95 T)}{8.47
↪   \times 10^{28} electrons/m^3}
The number of charge carriers per unit volume n is 8.47 \times 10^{28} electrons/m^3,
↪   which cancels out in the numerator and the denominator. Thus, we get: V = 0.95 V
The calculation is correct, and the final answer is indeed 0.95 V.
<judgment>correct</judgment>
```

