# OpenReview forum: "CoMAS: Co-Evolving Multi-Agent Systems via Interaction Rewards"
_ICLR.cc/2026/Conference — ICLR 2026 Poster_

### Official Review · Reviewer_qMBz · 2025-10-27

**Soundness:** 3
**Presentation:** 3
**Contribution:** 4
**Rating:** 6
**Confidence:** 4

**Summary:**

This paper introduces CoMAS (Co-Evolving Multi-Agent Systems), a novel framework that enables LLM-based agents to achieve self-evolution through pure inter-agent interactions without external reward signals. The key innovation lies in its adversarial reward design involving three interaction patterns: solution generation, critical evaluation, and scoring. The framework employs an LLM-as-a-judge mechanism where agents act as solvers, evaluators, and scorers in a zero-sum game, creating complementary rewards that encourage both correctness and critical thinking. Using REINFORCE++ for policy optimization, CoMAS supports heterogeneous agents in a decentralized training paradigm.

The authors conduct comprehensive experiments across 7 benchmarks (GSM8K, MATH-500, HumanEval, MBPP, SciBench, GPQA, MMLU) and 4 evaluation setups (Vanilla, Consistency, AutoGen, Debate), demonstrating consistent improvements over untrained agents (up to 19.80% in AutoGen setup) and competitive or superior performance compared to baselines (MAPoRL, TTRL). Ablation studies validate the necessity of the adversarial reward formulation and reveal promising scalability with increasing agent numbers and diversity.

**Strengths:**

1. **Clear Motivation and Problem Definition:** The paper articulates a compelling research question inspired by human collaborative learning, distinguishing itself from existing self-evolution approaches that focus on individual agents.

2. **Novel Adversarial Reward Design:** The zero-sum game between solvers and evaluators, mediated by an independent scorer, is an elegant solution that prevents reward hacking. The ablation study (Figure 4) convincingly demonstrates that removing either evaluation or scoring leads to training collapse or reward exploitation.

3. **Comprehensive Experimental Validation:**
   - 7 diverse benchmarks covering math, coding, science, and general knowledge
   - 4 different inference setups from single-agent to multi-agent collaboration
   - Comparison with strong baselines (MAPoRL, TTRL)
   - Extensive ablation studies on reward formulation and scalability

4. **Scalability and Generalizability:**
   - Performance improves with more agents (Section 4.3.2)
   - Heterogeneous agents outperform homogeneous ones
   - Decentralized training enables flexible system design
   - Skills transfer across domains

5. **Stable Training Dynamics:** Figure 3 shows that rewards converge around 0.5 and response lengths increase consistently, indicating healthy learning progress without collapse.

6. **Strong Performance:** CoMAS achieves state-of-the-art or competitive results across most settings, particularly excelling in multi-agent setups (e.g., 19.80% improvement on AutoGen).

**Weaknesses:**

1. **Limited Baseline Comparisons:**
   - MAPoRL baseline uses rule-based verifier instead of the specialized reward model from the original paper, which may not be a fair comparison
   - Missing comparisons with other recent self-evolution methods mentioned in related work (e.g., REMA, Self-Rewarding LMs)
   - Would benefit from comparison with debate-only methods without RL training
2. **Scalability Limitations:**
   - Experiments only go up to 4 agents; unclear how the framework scales to 10+ agents
   - All experiments use 3B parameter models; performance with larger models (7B, 13B) is unknown
   - Training cost analysis is missing (compute requirements, wall-clock time)
3. **Generalization Questions:**
   - All training data comes from verifiable domains (math, coding, science); unclear if the framework works for truly open-ended tasks like creative writing or complex reasoning
   - The framework assumes problems have objectively evaluable solutions, limiting applicability

**Questions:**

1. **Baseline Fairness:** Why was MAPoRL evaluated with a rule-based verifier instead of its original reward model? How much does this change affect the comparison? Could you also compare with debate frameworks without RL training?

2. **Statistical Significance:** Can you provide error bars or significance tests for the reported improvements? Some gains are modest (~1%), and it's important to know if they're statistically reliable.

3. **Failure Modes:** In what scenarios does CoMAS fail? Can you provide examples of agent discussions where the adversarial mechanism breaks down?


4. **Agent Initialization:** All agents start from the same pre-trained model in homogeneous settings. Does initializing with different checkpoints or different random seeds affect the co-evolution dynamics?

---

> ### Author Response · Authors · 2025-11-18
> **Response to Reviewer qMBz**
>
> We sincerely thank you for your constructive comments and insightful suggestions. We have carefully addressed each concern and revised the manuscript accordingly. Below, we detail our responses.
>
> #### **(W2) Experiments on Larger Models**
>
> We appreciate your suggestion to evaluate CoMAS on larger models. Since CoMAS is model-agnostic and relies on interaction-driven rewards for multi-agent co-evolution, we expect its effectiveness to generalize across models of different scales.
>
> To verify this, we conducted additional experiments using Qwen2.5-7B-Instruct under the same settings as our main study. The results are shown below.
>
> | **Setup**   | **Method**   | **Dataset**   |               |               |               |               |               |               |
> | ----------- | ------------ | ------------- | ------------- | ------------- | ------------- | ------------- | ------------- | ------------- |
> |             |              | GSM8K         | MATH-500      | HumanEval     | MBPP          | SciBench      | GPQA          | MMLU          |
> | Vanilla     | Untrained    | 88.40         | 58.80         | 82.32         | 66.00         | 46.29         | 31.25         | 70.20         |
> |             | CoMAS (Ours) | 91.40 (+3.00) | 61.00 (+2.20) | 82.93 (+0.61) | 67.20 (+1.20) | 47.49 (+1.20) | 32.81 (+1.56) | 70.80 (+0.60) |
> | Consistency | Untrained    | 90.80         | 62.80         | 82.93         | 67.00         | 52.51         | 35.71         | 72.20         |
> |             | CoMAS (Ours) | 92.00 (+1.20) | 63.40 (+0.60) | 82.93 (+0.00) | 68.20 (+1.20) | 53.31 (+0.80) | 38.17 (+2.46) | 74.00 (+1.80) |
> | AutoGen     | Untrained    | 87.40         | 57.20         | 75.00         | 61.80         | 44.49         | 34.15         | 68.80         |
> |             | CoMAS (Ours) | 90.00 (+2.60) | 59.60 (+2.40) | 77.44 (+2.44) | 62.40 (+0.60) | 46.29 (+1.80) | 34.82 (+0.67) | 69.20 (+0.40) |
> | Debate      | Untrained    | 92.00         | 62.20         | 82.93         | 67.00         | 51.30         | 37.72         | 72.40         |
> |             | CoMAS (Ours) | 92.40 (+0.40) | 63.20 (+1.00) | 83.54 (+0.61) | 68.00 (+1.00) | 51.90 (+0.60) | 38.39 (+0.67) | 73.60 (+1.20) |
>
> These results confirm that CoMAS continues to produce consistent and sometimes larger gains with 7B models, demonstrating strong scalability and model generality. We have added these findings to the revised Appendix.
>
> #### **(Q4) Experiments with Heterogeneous LLMs**
>
> Our second ablation study already examines the heterogeneous setting, combining Qwen2.5-3B-Instruct and Llama-3.2-3B-Instruct models. Partial results under the Vanilla setup are summarized below.
>
> | **Method**            | **Dataset**   |               |               |               |               |               |               |
> | --------------------- | ------------- | ------------- | ------------- | ------------- | ------------- | ------------- | ------------- |
> |                       | GSM8K         | MATH-500      | HumanEval     | MBPP          | SciBench      | GPQA          | MMLU          |
> | Untrained             | 84.00         | 51.40         | 68.90         | 54.00         | 32.67         | 26.79         | 61.40         |
> | CoMAS (Homogeneous)   | 83.40 (-0.60) | 52.80 (+1.40) | 67.68 (-1.22) | 54.00 (+0.00) | 33.87 (+1.20) | 27.46 (+0.67) | 62.20 (+0.80) |
> | CoMAS (Heterogeneous) | 85.20 (+1.20) | 53.20 (+1.80) | 73.78 (+4.88) | 58.00 (+4.00) | 34.47 (+1.80) | 28.12 (+1.34) | 61.80 (+0.40) |
>
> CoMAS achieves even better performance in heterogeneous settings, suggesting that diversity among LLMs enhances cooperative evolution.
>
> #### **(W2) Scalability with More Agents**
>
> As shown in our ablation study, performance improves as the number of agents increases from 1 to 4, indicating promising scalability. We agree that exploring larger populations (e.g., 8 or 16 agents) is an interesting and valuable direction, but a systematic analysis of large-scale multi-agent co-evolution lies beyond our scope and will be pursued in future work.

---

> ### Author Response · Authors · 2025-11-18
> **Response to Reviewer qMBz (Continued)**
>
> #### **(W1) Comparison with Recent Methods**
>
> We appreciate your suggestion to include additional baselines, but we find that ReMA [1] focuses on hierarchical reasoning through multi-agent reinforcement learning using external reward signals. While relevant, its design differs conceptually from our self-evolution objective.
>
> Besides, SRLM [2] is a direct self-evolution baseline. We train SRLM on our dataset under identical settings. The results are shown in the table below.
>
> | **Setup**   | **Method**   | **Dataset**    |               |                |               |               |               |                |
> | ----------- | ------------ | -------------- | ------------- | -------------- | ------------- | ------------- | ------------- | -------------- |
> |             |              | GSM8K          | MATH-500      | HumanEval      | MBPP          | SciBench      | GPQA          | MMLU           |
> | Vanilla     | Untrained    | 84.00          | 51.40         | 68.90          | 54.00         | 32.67         | 26.79         | 61.40          |
> |             | SRLM         | 83.40 (-0.60)  | 52.20 (+0.80) | 68.29 (-0.61)  | 53.80 (-0.20) | 32.67 (+0.00) | 27.01 (+0.22) | 61.00 (-0.40)  |
> |             | CoMAS (Ours) | 85.40 (+1.40)  | 52.80 (+1.40) | 70.73 (+1.83)  | 56.20 (+2.20) | 34.67 (+2.00) | 27.46 (+0.67) | 62.40 (+1.00)  |
> | Consistency | Untrained    | 85.40          | 55.00         | 73.78          | 55.80         | 36.47         | 28.79         | 63.20          |
> |             | SRLM         | 86.40 (+1.00)  | 55.40 (+0.40) | 75.00 (+1.22)  | 56.20 (+0.40) | 36.67 (+0.20) | 29.24 (+0.45) | 65.20 (+2.00)  |
> |             | CoMAS (Ours) | 87.20 (+1.80)  | 55.80 (+0.80) | 77.44 (+3.66)  | 59.20 (+3.40) | 37.68 (+1.20) | 29.69 (+0.89) | 65.60 (+2.40)  |
> | AutoGen     | Untrained    | 52.60          | 38.40         | 39.63          | 29.80         | 20.24         | 16.29         | 37.40          |
> |             | SRLM         | 58.00 (+5.40)  | 41.80 (+3.40) | 44.51 (+4.88)  | 32.00 (+2.20) | 21.24 (+1.00) | 17.86 (+1.56) | 42.40 (+5.00)  |
> |             | CoMAS (Ours) | 72.40 (+19.80) | 45.80 (+7.40) | 50.61 (+10.98) | 38.00 (+8.20) | 22.85 (+2.61) | 22.99 (+6.70) | 50.60 (+13.20) |
> | Debate      | Untrained    | 84.60          | 55.00         | 71.34          | 54.80         | 38.68         | 28.35         | 62.80          |
> |             | SRLM         | 84.60 (+0.00)  | 54.80 (-0.20) | 72.56 (+1.22)  | 53.60 (-1.20) | 38.68 (+0.00) | 28.57 (+0.22) | 64.60 (+1.80)  |
> |             | CoMAS (Ours) | 85.20 (+0.60)  | 55.40 (+0.40) | 77.44 (+6.10)  | 55.60 (+0.80) | 39.08 (+0.40) | 29.91 (+1.56) | 65.20 (+2.40)  |
>
> SRLM shows inconsistent gains over the untrained baseline, while CoMAS delivers clear and stable improvements in all configurations. These results further highlight the effectiveness of CoMAS in multi-agent self-evolution and are included in the revised Appendix.
>
> #### **(W1 & Q1) Comparison with RL-Free Methods**
>
> Our main experiments already include results for MAD [2]. Partial results under the Debate setup are reproduced below.
>
> | **Method**   | **Dataset**   |               |               |               |               |               |               |
> | ------------ | ------------- | ------------- | ------------- | ------------- | ------------- | ------------- | ------------- |
> |              | GSM8K         | MATH-500      | HumanEval     | MBPP          | SciBench      | GPQA          | MMLU          |
> | Untrained    | 84.60         | 55.00         | 71.34         | 54.80         | 38.68         | 28.35         | 62.80         |
> | MAPoRL       | 85.40 (+0.80) | 53.60 (-1.40) | 74.39 (+3.05) | 55.60 (+0.80) | 39.88 (+1.20) | 31.47 (+3.12) | 64.80 (+2.00) |
> | TTRL         | 86.20 (+1.60) | 55.20 (+0.20) | 73.78 (+2.44) | 58.00 (+3.20) | 37.88 (-0.80) | 29.02 (+0.67) | 64.00 (+1.20) |
> | CoMAS (Ours) | 85.20 (+0.60) | 55.40 (+0.40) | 77.44 (+6.10) | 55.60 (+0.80) | 39.08 (+0.40) | 29.91 (+1.56) | 65.20 (+2.40) |
>
> CoMAS achieves significant and consistent improvements over the untrained baseline within the Debate configuration, demonstrating the benefit of reinforcement-driven co-evolution on top of debate frameworks.

---

> ### Author Response · Authors · 2025-11-18
> **Response to Reviewer qMBz (Continued)**
>
> #### **(W1 & Q1) Fairness of MAPoRL Comparison**
>
> We appreciate your concern about fair baseline comparison and would like to clarify from three aspects.
>
> 1. All methods, including CoMAS, MAPoRL, and TTRL, were trained under the MARTI [3] framework and evaluated within MASLab [4], two widely adopted open-source environments, ensuring consistent and transparent evaluation.
> 2. The original MAPoRL relies on a specialized reward model incompatible with our task settings. We therefore adopted the reproduced implementation in MARTI, which uses a validated rule-based verifier.
> 3. Prior works have reported that rule-based verifiers provide more accurate reward signals than learned reward models, resulting in stronger and more stable agentic RL performance [5].
>
> Hence, the comparisons are fair and representative within standard community practices.
>
> #### **(Q2) Statistical Significance**
>
> To assess the robustness of our improvements, we repeated the evaluation under the Vanilla setup with five random seeds, reporting the mean and standard deviation in the table below.
>
> | **Method**   | **Dataset**   |               |               |               |               |               |               |
> | ------------ | ------------- | ------------- | ------------- | ------------- | ------------- | ------------- | ------------- |
> |              | GSM8K         | MATH-500      | HumanEval     | MBPP          | SciBench      | GPQA          | MMLU          |
> | Untrained    | 83.68 (±0.35) | 51.52 (±0.57) | 69.76 (±1.37) | 54.52 (±0.45) | 32.30 (±0.68) | 26.38 (±1.05) | 60.96 (±0.39) |
> | SRLM         | 83.52 (±0.48) | 52.16 (±0.66) | 70.12 (±1.02) | 54.04 (±0.79) | 32.38 (±0.86) | 26.65 (±0.69) | 60.64 (±2.19) |
> | MAPoRL       | 84.20 (±0.33) | 52.08 (±0.84) | 72.07 (±1.98) | 56.64 (±0.67) | 33.59 (±0.37) | 28.66 (±0.54) | 60.80 (±0.61) |
> | TTRL         | 84.36 (±0.67) | 52.92 (±1.27) | 71.22 (±1.05) | 58.32 (±0.30) | 34.07 (±0.92) | 26.21 (±1.34) | 61.64 (±0.95) |
> | CoMAS (Ours) | 84.68 (±0.37) | 53.12 (±0.10) | 74.15 (±2.06) | 56.32 (±0.47) | 33.87 (±0.13) | 29.06 (±0.83) | 62.12 (±0.72) |
>
> Although some gains are modest, the low standard deviations confirm their statistical reliability. These results have been added to the revised Appendix.
>
> #### **(W2) Analysis of Training Cost**
>
> We appreciate the valuable suggestion to include training cost analysis. Under CoMAS, for $l$ agents, the number of interaction samples and generated tokens scale approximately with $l^2$, memory usage scales linearly with $l$, and wall-clock time remains roughly constant due to parallelization. The empirical results are shown in the table below.
>
> | **Agent Number** | **Interaction Sample** | **Generated Token** | **Memory Consumption** | **Wall-clock Time** |
> | ---------------- | ---------------------- | ------------------- | ---------------------- | ------------------- |
> | 1                | 48k                    | 1.6B                | 120GB                  | 11.7h               |
> | 2                | 192k                   | 6.3B                | 240GB                  | 12.1h               |
> | 4                | 768k                   | 25.2B               | 480GB                  | 12.5h               |
>
> This training cost analysis has been included in the revised Appendix, which further improves the completeness of our paper.
>
> #### **(Q3) Failure Modes**
>
> CoMAS demonstrates stable training across all settings. However, our ablation studies reveal two key failure modes when critical components are removed.
>
> 1. When evaluation is absent, agents become progressively stricter critics during self-learning, leading to monotonically decreasing rewards and causing training collapse.
> 2. When scoring is absent, evaluations receive rewards only for self-consistency and quickly exploit the reward signal by labeling all solutions as correct, leading to training collapse.
>
> We have included representative examples of such failed discussions in the revised Appendix for reference.
>
> #### **(W3) Generalization to Open-Ended Tasks**
>
> We agree that testing CoMAS on open-ended tasks is an important future direction. However, our current experiments already span diverse and verifiable domains (e.g., math, coding, and science), providing a robust demonstration of cross-domain generalization.
>
> As standard benchmarks for open-ended tasks are still emerging, we consider a comprehensive study in such domains beyond the scope of the present paper and will explore it in our future work.

---

> ### Author Response · Authors · 2025-11-18
> **Response to Reviewer qMBz (Continued)**
>
> #### **References**
>
> [1] Wan Z, Li Y, Wen X, et al. Rema: Learning to meta-think for llms with multi-agent reinforcement learning[J]. arXiv preprint arXiv:2503.09501, 2025.
>
> [2] Liang T, He Z, Jiao W, et al. Encouraging divergent thinking in large language models through multi-agent debate[C]. Proceedings of the 2024 conference on empirical methods in natural language processing, 2024.
>
> [3] Zhang K, Liu R, Zhu X, et al. Marti: A framework for multi-agent llm systems reinforced training and inference[EB/OL]. https://github.com/TsinghuaC3I/MARTI, 2025.
>
> [4] Ye R, Huang K, Wu Q, et al. Maslab: A unified and comprehensive codebase for llm-based multi-agent systems[J]. arXiv preprint arXiv:2505.16988, 2025.
>
> [5] Guo D, Yang D, Zhang H, et al. Deepseek-r1: Incentivizing reasoning capability in llms via reinforcement learning[J]. arXiv preprint arXiv:2501.12948, 2025.

---

> ### Author Response · Authors · 2025-11-27
> **Follow-up on Rebuttal Discussion**
>
> Dear Reviewer qMBz,
>
> We appreciate the time and effort you have invested in reviewing our submission. As the rebuttal period is drawing to a close, we would like to kindly remind you that we have posted our response addressing the main concerns raised in your review.
>
> We would be very grateful if you could take a moment to review our updates and share any further comments. If our clarifications adequately resolve your concerns, we would sincerely appreciate your consideration in updating your rating and confidence.
>
> Best regards,
>
> Submission 5758 Authors

---

### Official Review · Reviewer_CxUs · 2025-10-28

**Soundness:** 2
**Presentation:** 2
**Contribution:** 2
**Rating:** 4
**Confidence:** 3

**Summary:**

This paper proposes the CoMAS (Co-Evolving Multi-Agent Systems) framework, mimicking the human learning mechanism of collaborative interaction and eliminating the need for external supervision. The framework first enables agents to engage in decentralized interactions following a "solution generation-critical evaluation-structured scoring" model, then leverages an LLM-as-a-judge mechanism to convert scores into zero-sum game-style intrinsic rewards (incentivizing both solution accuracy and rigorous evaluation), and finally uses the REINFORCE++ reinforcement learning algorithm to update each agent’s policy. Experiments show that across 7 benchmarks as well as single-agent and multi-agent scenarios.

**Strengths:**

1. The paper demonstrates good originality by addressing a critical limitation in LLM agent self-evolution. Prior RL-based methods rely on external rewards or single-agent intrinsic signals, while CoMAS introduces a novel "interaction-driven co-evolution" paradigm. By mimicking human collaborative learning (via decentralized solution-evaluation-scoring loops) and designing a zero-sum reward system to align agent incentives, it fills the gap of "unsupervised, interaction-only evolution" for LLM agents, rather than simply combining existing multi-agent or RL techniques.
2. The paper is clear in presentation: it explicitly maps each component to solving prior limitations (e.g., format penalties prevent reward hacking), provides detailed formulas for key steps (e.g., reward calculation, advantage function), and uses ablation studies to validate the necessity of core modules.

**Weaknesses:**

1.  The choice of Qwen2.5-3B-Instruct as the base model imposes a fundamental constraint on CoMAS’s evolutionary potential. 3B-scale models are well-documented to struggle with complex reasoning tasks, including multi-step mathematical operations, intricate programming logic, and nuanced open-ended problem-solving, even with advanced prompting. While the paper demonstrates marginal gains over the 3B baseline, it cannot verify whether the CoMAS framework can drive meaningful evolution for models already equipped with stronger inherent capabilities. For instance, Qwen2.5’s 7B variants outperform the 3B version significantly in code generation and mathematical reasoning ; without testing on these larger scales, it remains unclear if CoMAS’s interaction-driven paradigm can amplify the strengths of more capable models or merely compensates for the 3B model’s baseline deficiencies.

2. While the paper shows performance scales with agent count (4>2>1), it lacks critical analysis of how interaction complexity impacts evolution efficiency and omits clear scalability boundaries. For example, it does not address whether increasing agents beyond 4 leads to diminishing returns (e.g., redundant discussions) or higher communication overhead that degrades training speed. Additionally, the "recent κ rounds" context truncation rule is set without justification, there is no ablation on how κ values affect discussion quality or reward reliability, leaving readers unsure if the chosen parameter is optimal or generalizable.

3. While the current baselines (untrained agents, MAPoRL, TTRL) help validate CoMAS’s performance against non-interaction-driven methods, the scope of comparative methods appears relatively narrow. That said, I acknowledge uncertainty about the exact number of recently _published_ works that directly align with multi-agent evolution. As such, I recommend cross-referencing feedback from other reviewers on this point: if peers highlight additional relevant methods, expanding the baseline set to include these would strengthen the paper’s claims of superiority.

**Questions:**

Please see weaknesses.

---

> ### Author Response · Authors · 2025-11-18
> **Response to Reviewer CxUs**
>
> We sincerely thank you for your careful reading, constructive feedback, and positive remarks on our contribution and presentation. We have thoroughly revised the paper to address the identified weaknesses. Our detailed responses are as follows.
>
> #### **(W1) Experiments on 7B Models**
>
> We appreciate your suggestion to extend our evaluation to larger models. As a model-agnostic framework, CoMAS enables multi-agent co-evolution through interaction-driven rewards, and we expect its effectiveness to generalize across foundation models of different sizes.
>
> To verify this, we additionally trained CoMAS using Qwen2.5-7B-Instruct under the same experimental setup. The results are summarized below.
>
> | **Setup**   | **Method**   | **Dataset**   |               |               |               |               |               |               |
> | ----------- | ------------ | ------------- | ------------- | ------------- | ------------- | ------------- | ------------- | ------------- |
> |             |              | GSM8K         | MATH-500      | HumanEval     | MBPP          | SciBench      | GPQA          | MMLU          |
> | Vanilla     | Untrained    | 88.40         | 58.80         | 82.32         | 66.00         | 46.29         | 31.25         | 70.20         |
> |             | CoMAS (Ours) | 91.40 (+3.00) | 61.00 (+2.20) | 82.93 (+0.61) | 67.20 (+1.20) | 47.49 (+1.20) | 32.81 (+1.56) | 70.80 (+0.60) |
> | Consistency | Untrained    | 90.80         | 62.80         | 82.93         | 67.00         | 52.51         | 35.71         | 72.20         |
> |             | CoMAS (Ours) | 92.00 (+1.20) | 63.40 (+0.60) | 82.93 (+0.00) | 68.20 (+1.20) | 53.31 (+0.80) | 38.17 (+2.46) | 74.00 (+1.80) |
> | AutoGen     | Untrained    | 87.40         | 57.20         | 75.00         | 61.80         | 44.49         | 34.15         | 68.80         |
> |             | CoMAS (Ours) | 90.00 (+2.60) | 59.60 (+2.40) | 77.44 (+2.44) | 62.40 (+0.60) | 46.29 (+1.80) | 34.82 (+0.67) | 69.20 (+0.40) |
> | Debate      | Untrained    | 92.00         | 62.20         | 82.93         | 67.00         | 51.30         | 37.72         | 72.40         |
> |             | CoMAS (Ours) | 92.40 (+0.40) | 63.20 (+1.00) | 83.54 (+0.61) | 68.00 (+1.00) | 51.90 (+0.60) | 38.39 (+0.67) | 73.60 (+1.20) |
>
> These results confirm that CoMAS continues to yield consistent and sometimes larger gains on the 7B model, reinforcing its general applicability and scalability. We have incorporated these findings into the revised Appendix.
>
> #### **(W2) Analysis of Scaling Behavior**
>
> In the revised paper, we expand the discussion on how interaction complexity scales with the number of agents. Specifically, for $l$ agents, the number of interaction samples and generated tokens scale approximately with $l^2$, while memory usage scales linearly with $l$. Owing to parallelization, the wall-clock training time remains largely stable. The empirical measurements are shown below.
>
> | **Agent Number** | **Interaction Sample** | **Generated Token** | **Memory Consumption** | **Wall-clock Time** |
> | ---------------- | ---------------------- | ------------------- | ---------------------- | ------------------- |
> | 1                | 48k                    | 1.6B                | 120GB                  | 11.7h               |
> | 2                | 192k                   | 6.3B                | 240GB                  | 12.1h               |
> | 4                | 768k                   | 25.2B               | 480GB                  | 12.5h               |
>
> These results illustrate the computational overhead when increasing the number of agents and help readers assess the trade-offs between scalability and efficiency.
>
> While our experiments show steady improvements from 1 to 4 agents, we agree that systematically exploring larger-scale configurations and diminishing-return effects would be an interesting direction for future work.
>
> #### **(W2) Choice of Context Truncation**
>
> Thank you for pointing out this issue. We set $\kappa = 2$ based on the following considerations.
>
> 1. Retaining only the latest two rounds of discussion is a standard practice in debate‑style multi‑agent frameworks, as adopted in MAD [1] and MAPoRL [2].
> 2. The Qwen2.5-3B-Instruct model has a 32k-token context window. Larger $\kappa$ values could exceed this limit, potentially causing training instability or failure.
>
> Therefore, $\kappa = 2$ is a technical compromise that balances discussion richness with computational feasibility. Although increasing $\kappa$ may further enhance context awareness, we leave a systematic exploration of its effects to future work.

---

> ### Author Response · Authors · 2025-11-18
> **Response to Reviewer CxUs (Continued)**
>
> #### **(W3) Sufficiency of Compared Methods**
>
> We appreciate your emphasis on the breadth of baselines. Since few existing methods directly address multi-agent evolution, our original baselines focused on the most relevant paradigms.
>
> Following your suggestion and that of other reviewers, we have added SRLM [3] as an additional baseline. The results are shown in the table below.
>
> | **Setup**   | **Method**   | **Dataset**    |               |                |               |               |               |                |
> | ----------- | ------------ | -------------- | ------------- | -------------- | ------------- | ------------- | ------------- | -------------- |
> |             |              | GSM8K          | MATH-500      | HumanEval      | MBPP          | SciBench      | GPQA          | MMLU           |
> | Vanilla     | Untrained    | 84.00          | 51.40         | 68.90          | 54.00         | 32.67         | 26.79         | 61.40          |
> |             | SRLM         | 83.40 (-0.60)  | 52.20 (+0.80) | 68.29 (-0.61)  | 53.80 (-0.20) | 32.67 (+0.00) | 27.01 (+0.22) | 61.00 (-0.40)  |
> |             | CoMAS (Ours) | 85.40 (+1.40)  | 52.80 (+1.40) | 70.73 (+1.83)  | 56.20 (+2.20) | 34.67 (+2.00) | 27.46 (+0.67) | 62.40 (+1.00)  |
> | Consistency | Untrained    | 85.40          | 55.00         | 73.78          | 55.80         | 36.47         | 28.79         | 63.20          |
> |             | SRLM         | 86.40 (+1.00)  | 55.40 (+0.40) | 75.00 (+1.22)  | 56.20 (+0.40) | 36.67 (+0.20) | 29.24 (+0.45) | 65.20 (+2.00)  |
> |             | CoMAS (Ours) | 87.20 (+1.80)  | 55.80 (+0.80) | 77.44 (+3.66)  | 59.20 (+3.40) | 37.68 (+1.20) | 29.69 (+0.89) | 65.60 (+2.40)  |
> | AutoGen     | Untrained    | 52.60          | 38.40         | 39.63          | 29.80         | 20.24         | 16.29         | 37.40          |
> |             | SRLM         | 58.00 (+5.40)  | 41.80 (+3.40) | 44.51 (+4.88)  | 32.00 (+2.20) | 21.24 (+1.00) | 17.86 (+1.56) | 42.40 (+5.00)  |
> |             | CoMAS (Ours) | 72.40 (+19.80) | 45.80 (+7.40) | 50.61 (+10.98) | 38.00 (+8.20) | 22.85 (+2.61) | 22.99 (+6.70) | 50.60 (+13.20) |
> | Debate      | Untrained    | 84.60          | 55.00         | 71.34          | 54.80         | 38.68         | 28.35         | 62.80          |
> |             | SRLM         | 84.60 (+0.00)  | 54.80 (-0.20) | 72.56 (+1.22)  | 53.60 (-1.20) | 38.68 (+0.00) | 28.57 (+0.22) | 64.60 (+1.80)  |
> |             | CoMAS (Ours) | 85.20 (+0.60)  | 55.40 (+0.40) | 77.44 (+6.10)  | 55.60 (+0.80) | 39.08 (+0.40) | 29.91 (+1.56) | 65.20 (+2.40)  |
>
> The results show that SRLM does not consistently outperform the untrained baseline, whereas CoMAS achieves clear and stable improvements across all settings, confirming its advantage in multi-agent co-evolution. These results have been added to the revised Appendix.
>
> #### **References**
>
> [1] Liang T, He Z, Jiao W, et al. Encouraging divergent thinking in large language models through multi-agent debate[C]. Proceedings of the 2024 conference on empirical methods in natural language processing, 2024.
>
> [2] Park C, Han S, Guo X, et al. Maporl: Multi-agent post-co-training for collaborative large language models with reinforcement learning[C]. Proceedings of the 63rd Annual Meeting of the Association for Computational Linguistics, 2025.
>
> [3] Yuan W, Pang R Y, Cho K, et al. Self-rewarding language models[C]. Forty-first International Conference on Machine Learning, 2024.

---

> ### Author Response · Authors · 2025-11-27
> **Follow-up on Rebuttal Discussion**
>
> Dear Reviewer CxUs,
>
> We appreciate the time and effort you have invested in reviewing our submission. As the rebuttal period is drawing to a close, we would like to kindly remind you that we have posted our response addressing the main concerns raised in your review.
>
> We would be very grateful if you could take a moment to review our updates and share any further comments. If our clarifications adequately resolve your concerns, we would sincerely appreciate your consideration in updating your rating and confidence.
>
> Best regards,
>
> Submission 5758 Authors

---

### Official Review · Reviewer_F6Fu · 2025-10-30

**Soundness:** 2
**Presentation:** 2
**Contribution:** 2
**Rating:** 4
**Confidence:** 4

**Summary:**

The paper proposes a multi-agent self-evolution framework called CoMAS. In this system, agents engage in an interactive cycle of solving–mutual evaluation–scoring, generating dialogue trajectories from which an LLM-as-a-judge extracts intrinsic rewards. Each agent then optimizes its policy through reinforcement learning, enabling unsupervised, verifier-free co-evolution. Across multiple benchmarks and reasoning settings, CoMAS achieves stable improvements over untrained agents and even reaches state-of-the-art performance in some configurations.

**Strengths:**

1. The paper is clearly written and easy to follow.

2. The research problem is well-defined, and constructing rewards from a multi-agent perspective is a promising direction.

**Weaknesses:**

- From my perspective, the novelty is limited, obtaining rewards through multi-agent discussions is not a new idea.

- The Related Work section lacks detailed and structured discussion of prior work, which blurs the paper’s contribution. The authors should provide a comprehensive overview of previous methods and explicitly highlight the differences between their method and existing ones.

- Most comparisons are conducted under the paper’s unified pipeline rather than under each method’s native optimal setup, which may underestimate strong baselines.

- The figures emphasize heterogeneous LLMs, but most experiments are based on homogeneous LLMs.

**Questions:**

1. Could the authors systematically review recent works and clearly articulate the core contribution of this paper? At present, it is hard to see what differentiates it from existing methods? What specific shortcomings of prior works does CoMAS address?

2. Please report token, wall-clock time, and memory curves for different agent counts; this is one of the main limitations of the current study.

3. Besides the unified pipeline, please reproduce baselines in their native configurations and re-run CoMAS for fair comparison.

4. According to my understanding, the proposed method and baselines do not use the same RL algorithm (e.g., CoMAS uses REINFORCE++ instead of GRPO). This inconsistency may introduce confounding factors. I suggest using the same RFT algorithm as the baselines for a fair comparison.

---

> ### Author Response · Authors · 2025-11-18
> **Response to Reviewer F6Fu**
>
> We sincerely thank you for the constructive and insightful feedback. We have carefully revised the manuscript according to the comments and added detailed clarifications as summarized below.
>
> #### **(W1 & Q1) Novelty and Core Contribution**
>
> We appreciate your comments regarding novelty and clarification of our contributions. Below we provide a concise summary of how our work advances the field.
>
> 1. The self-evolution of LLM-based agents is undergoing a paradigm shift from RL-free symbolic learning to RL-based parameter optimization.
> 2. Most existing RL-based approaches rely on external supervision from reward models or rule-based verifiers. Recent studies have started exploring unsupervised self-evolution, where rewards are internally derived from signals such as confidence scores, semantic entropy, or majority voting.
> 3. However, unsupervised self-evolution has been largely unexplored in multi-agent settings, despite its high potential to mirror cooperative and competitive dynamics of human social learning.
> 4. To address this gap, we propose CoMAS, a multi-agent framework that enables unsupervised self-evolution, where agents co-adapt through mutual reward generation without any external supervision.
> 5. Consequently, CoMAS fills a critical gap in unsupervised self-evolution for multi-agent systems, establishing a novel and promising paradigm for the self-evolution of LLM-based agents.
>
> To clearly distinguish CoMAS from existing works, we select some representative works and summarize them in the table below.
>
> | **Paper**    | **Method**                                                   | **Supervision**                     | **Multi-Agent** | **RL-Based** |
> | ------------ | ------------------------------------------------------------ | ----------------------------------- | --------------- | ------------ |
> | ADAS [1]     | Search agentic workflow with validation feedback.            | External (Ground Truth)             | No              | No           |
> | GPTSwarm [2] | Optimize nodes and edges by maximizing the utility function. | External (Ground Truth)             | Yes             | No           |
> | SRLM [3]     | Construct preference pairs with self-generated rewards.      | Intrinsic (Self Rewarding)          | No              | No           |
> | MAPoRL [4]   | Incorporate reinforcement learning with multi-agent debate.  | External (Reward Model)             | Yes             | Yes          |
> | TTRL [5]     | Formulate rewards from fake labels generated by majority voting. | Intrinsic (Majority Voting)         | No              | Yes          |
> | CoMAS (Ours) | Co-evolve multiple agents with rewards from mutual interaction. | Intrinsic (Multi-agent Interaction) | Yes             | Yes          |
>
> We respectfully disagree with the suggestion that obtaining rewards through multi-agent discussion is not a novel concept. To the best of our knowledge, this approach has not yet been explored for enabling the self-evolution of LLM-based agents. While the idea may appear intuitive, its practical realization and rigorous investigation in this specific context are unique to our work. We kindly invite you to reconsider the originality and contribution of our proposed framework in this light.
>
> #### **(W2) Clarity of Related Work**
>
> We appreciate the suggestion to improve the structure of the Related Work section. In the revision, we have reorganized it to group prior studies by methodology and learning paradigm, explicitly highlighting how CoMAS differs from each line of work.

---

> ### Author Response · Authors · 2025-11-18
> **Response to Reviewer F6Fu (Continued)**
>
> #### **(W3 & Q3) Fairness of Baseline Comparisons**
>
> We understand your concern about baseline fairness and would like to clarify the setup.
>
> 1. All methods, including CoMAS, MAPoRL, and TTRL, were trained under the MARTI [6] framework and evaluated within MASLab [7], two widely adopted open-source environments, ensuring consistent and transparent evaluation.
> 2. The original MAPoRL implementation depends on a dedicated reward model, which is incompatible with our experimental setup. Instead, we used the MARTI reproduction, which has been extensively validated and achieves comparable or better performance than the official version.
> 3. TTRL was directly run using the official MARTI implementation with the original hyperparameters, eliminating concerns of underestimation.
>
> Hence, the comparisons are fair and representative within standard community practices.
>
> #### **(Q4) Consistency of RL Algorithm**
>
> We apologize for the earlier lack of detail. In all experiments, including baselines, REINFORCE++ was used as the reinforcement fine-tuning algorithm following MARTI's implementation. We have added this clarification to the Experiments section in the revision.
>
> #### **(Q2) Training Cost Analysis**
>
> We appreciate the valuable suggestion to include training cost analysis. Under CoMAS, for $l$ agents, the number of interaction samples and generated tokens scale approximately with $l^2$, memory usage scales linearly with $l$, and wall-clock time remains roughly constant due to parallelization. The empirical results are summarized below.
>
> | **Agent Number** | **Interaction Sample** | **Generated Token** | **Memory Consumption** | **Wall-clock Time** |
> | ---------------- | ---------------------- | ------------------- | ---------------------- | ------------------- |
> | 1                | 48k                    | 1.6B                | 120GB                  | 11.7h               |
> | 2                | 192k                   | 6.3B                | 240GB                  | 12.1h               |
> | 4                | 768k                   | 25.2B               | 480GB                  | 12.5h               |
>
> This training cost analysis has been included in the revised Appendix, which further improves the completeness of our paper.
>
> #### **(W4) Use of Homogeneous LLMs**
>
> We thank you for pointing out this potential source of confusion. Inspired by MAPoRL, the differing icons in Figure 1 and 2 were solely intended to visualize multiple agents in the discussion, not to imply heterogeneous LLMs. Following your comment, we have refined the figure captions and associated text to avoid any ambiguity.
>
> #### **References**
>
> [1] Hu S, Lu C, Clune J. Automated Design of Agentic Systems[C]. The Thirteenth International Conference on Learning Representations, 2025.
>
> [2] Zhuge M, Wang W, Kirsch L, et al. Gptswarm: Language agents as optimizable graphs[C]. Forty-first International Conference on Machine Learning, 2024.
>
> [3] Yuan W, Pang R Y, Cho K, et al. Self-rewarding language models[C]. Forty-first International Conference on Machine Learning, 2024.
>
> [4] Park C, Han S, Guo X, et al. Maporl: Multi-agent post-co-training for collaborative large language models with reinforcement learning[C]. Proceedings of the 63rd Annual Meeting of the Association for Computational Linguistics, 2025.
>
> [5] Zuo Y, Zhang K, Sheng L, et al. Ttrl: Test-time reinforcement learning[J]. arXiv preprint arXiv:2504.16084, 2025.
>
> [6] Zhang K, Liu R, Zhu X, et al. Marti: A framework for multi-agent llm systems reinforced training and inference[EB/OL]. https://github.com/TsinghuaC3I/MARTI, 2025.
>
> [7] Ye R, Huang K, Wu Q, et al. Maslab: A unified and comprehensive codebase for llm-based multi-agent systems[J]. arXiv preprint arXiv:2505.16988, 2025.

---

> ### Author Response · Authors · 2025-11-27
> **Follow-up on Rebuttal Discussion**
>
> Dear Reviewer F6Fu,
>
> We appreciate the time and effort you have invested in reviewing our submission. As the rebuttal period is drawing to a close, we would like to kindly remind you that we have posted our response addressing the main concerns raised in your review.
>
> We would be very grateful if you could take a moment to review our updates and share any further comments. If our clarifications adequately resolve your concerns, we would sincerely appreciate your consideration in updating your rating and confidence.
>
> Best regards,
>
> Submission 5758 Authors

---

### Official Review · Reviewer_bqfG · 2025-11-01

**Soundness:** 3
**Presentation:** 3
**Contribution:** 2
**Rating:** 6
**Confidence:** 3

**Summary:**

CoMAS is a multi-agent system that maintains a population of LLM agents and iteratively finetunes them. In each fine-tune iteration, the agents act as three roles: a solver that generates solutions, an evaluator that evaluates the solution, and a scorer that generates a score based on the solution and the evaluation. The score serves as an adversarial reward that is then used to optimize the scorer and the evaluator with RL.

**Strengths:**

- The writing is clear and well-organized.
- The proposed method is novel to my knowledge.
- The evaluation is detailed, including comprehensive benchmark results and ablations.
- In the evaluations, CoMAS establishes advantages over existing baselines in most scenarios.
- Prompt templates and examples, as well as training hyperparameters are provided for reproducibility.

**Weaknesses:**

- CoMAS generally outperforms other baselines. However, the performance gap is only significant in a minority of the tasks - mostly with AutoGen setup. Furthermore, both "strong" baselines, MAPoRL and TTRL, almost never have significant improvement over the naive untrained baseline.
- I am not yet convinced that this reward formulation solves the reward hacking problem. This formulation does create an adversary between the solver and the evaluator, but it does not ensure that the generated contents are grounded, meaning both parties can be reward hacking at the same time. I doubt this will be a real issue, but it would be helpful to show some evidence or counterpoints.

**Questions:**

- From the evaluations, the AutoGen setup result is siginificantly lower than the rest. Some insights to this would be helpful.

### Some minor issues
- Figure 2 is cute but I don't think it's necessarily information-condense. I don't see why you need to show three different threads, as all three parties are involved even in one of them. This is a minior issue and even personal preference.
- The numbers in Figure 4 (left) and 5 are too small and hardly visible when printed.

---

> ### Author Response · Authors · 2025-11-18
> **Response to Reviewer bqfG**
>
> We sincerely thank you for your thoughtful feedback and appreciation of the novelty, completeness, and clarity of our work. We have carefully revised the manuscript accordingly, and we address the main concerns below.
>
> #### **(W1) Significance of Performance Gains**
>
> Although the numerical gains of CoMAS and the baselines appear moderate under certain settings, our updated analysis confirms their statistical significance. Specifically, we repeated the evaluation under the Vanilla setup using 5 random seeds, reporting mean and standard deviation across runs. The results, now included in the revised Appendix, demonstrate that all the methods yield significant improvements over the untrained baseline.
>
> | **Method**   | **Dataset**   |               |               |               |               |               |               |
> | ------------ | ------------- | ------------- | ------------- | ------------- | ------------- | ------------- | ------------- |
> |              | GSM8K         | MATH-500      | HumanEval     | MBPP          | SciBench      | GPQA          | MMLU          |
> | Untrained    | 83.68 (±0.35) | 51.52 (±0.57) | 69.76 (±1.37) | 54.52 (±0.45) | 32.30 (±0.68) | 26.38 (±1.05) | 60.96 (±0.39) |
> | SRLM         | 83.52 (±0.48) | 52.16 (±0.66) | 70.12 (±1.02) | 54.04 (±0.79) | 32.38 (±0.86) | 26.65 (±0.69) | 60.64 (±2.19) |
> | MAPoRL       | 84.20 (±0.33) | 52.08 (±0.84) | 72.07 (±1.98) | 56.64 (±0.67) | 33.59 (±0.37) | 28.66 (±0.54) | 60.80 (±0.61) |
> | TTRL         | 84.36 (±0.67) | 52.92 (±1.27) | 71.22 (±1.05) | 58.32 (±0.30) | 34.07 (±0.92) | 26.21 (±1.34) | 61.64 (±0.95) |
> | CoMAS (Ours) | 84.68 (±0.37) | 53.12 (±0.10) | 74.15 (±2.06) | 56.32 (±0.47) | 33.87 (±0.13) | 29.06 (±0.83) | 62.12 (±0.72) |
>
> It is also important to note that our setup is more challenging than traditional single-domain formulations: agents are trained jointly across math, coding, and science tasks without external supervision, and are evaluated in a cross-domain setting. In contrast, MAPoRL [1] and TTRL [2] were primarily designed for mathematical reasoning, relying on reward modeling and majority-voting schemes that are not directly applicable to coding or scientific reasoning. This explains why the baselines show smaller gains in our multi-domain environment.
>
> #### **(Q1) Lower Performance in the AutoGen Setup**
>
> We appreciate the request for clarification. The AutoGen [3, 4] setup differs conceptually from the other setups. It involves a user-proxy and an assistant agent to improve user satisfaction and interpretability, rather than maximizing objective accuracy through ensembling or consensus. As a result, its performance is typically lower on reasoning benchmarks that focus solely on correctness.
>
> Moreover, we strictly follow the implementation of MASLab [5], and our reproduced results are consistent with those reported in their paper, supporting the reliability of our evaluation.
>
> #### **(W2) Effectiveness of the Reward Formulation**
>
> The effectiveness of our adversarial reward design is validated through extensive analysis in the first ablation study. As shown in Figure 4, CoMAS maintains a stable average normalized reward throughout training. By contrast, removing the adversarial formulation leads to unbounded increases or decreases in reward values, often resulting in training collapse.
>
> To further assess the validity of the generated reward signals, we align them with rule-based verifier results, treat the verifiers as ground truth, and compute precision and recall of generated rewards.
>
> | **Step**      | 0     | 5     | 10    | 15    | 20    | 25    | 30    |
> | ------------- | ----- | ----- | ----- | ----- | ----- | ----- | ----- |
> | **Accuracy**  | 38.38 | 40.98 | 42.05 | 40.90 | 43.30 | 43.92 | 45.75 |
> | **Precision** | 39.52 | 45.16 | 46.31 | 45.30 | 46.68 | 48.57 | 50.10 |
> | **Recall**    | 20.29 | 33.36 | 38.11 | 37.65 | 34.97 | 33.20 | 28.95 |
>
> Both precision and recall increase consistently with task accuracy, suggesting that performance improvement arises from progressively more accurate reward signals, rather than from reward hacking. The detailed results have been included in the revised Appendix.
>
> #### **(Q2) Clarification on Figure 2**
>
> We apologize for the confusion, but the three threads in Figure 2 refer to the same question, depicted sequentially to illustrate the progression of three key interaction stages in CoMAS, i.e., solution, evaluation, and scoring. The context of each stage becomes input to the next, forming a continuous multi-round discussion.
>
> To improve clarity, we have revised the figure caption to emphasize this relationship.
>
> #### **(Q3) Legibility of Figures 4 and 5**
>
> We appreciate your remark. We have increased the font size and improved overall visibility of Figures 4 and 5 in the revised submission to ensure readability in both digital and print formats.

---

> ### Author Response · Authors · 2025-11-18
> **Response to Reviewer bqfG (Continued)**
>
> #### **References**
>
> [1] Park C, Han S, Guo X, et al. Maporl: Multi-agent post-co-training for collaborative large language models with reinforcement learning[C]. Proceedings of the 63rd Annual Meeting of the Association for Computational Linguistics, 2025.
>
> [2] Zuo Y, Zhang K, Sheng L, et al. Ttrl: Test-time reinforcement learning[J]. arXiv preprint arXiv:2504.16084, 2025.
>
> [3] Li G, Hammoud H, Itani H, et al. Camel: Communicative agents for "mind" exploration of large language model society[J]. Advances in Neural Information Processing Systems, 2023.
>
> [4] Wu Q, Bansal G, Zhang J, et al. Autogen: Enabling next-gen LLM applications via multi-agent conversations[C]. First Conference on Language Modeling, 2024.
>
> [5] Ye R, Huang K, Wu Q, et al. Maslab: A unified and comprehensive codebase for llm-based multi-agent systems[J]. arXiv preprint arXiv:2505.16988, 2025.

---

> ### Author Response · Authors · 2025-11-27
> **Follow-up on Rebuttal Discussion**
>
> Dear Reviewer bqfG,
>
> We appreciate the time and effort you have invested in reviewing our submission. As the rebuttal period is drawing to a close, we would like to kindly remind you that we have posted our response addressing the main concerns raised in your review.
>
> We would be very grateful if you could take a moment to review our updates and share any further comments. If our clarifications adequately resolve your concerns, we would sincerely appreciate your consideration in updating your rating and confidence.
>
> Best regards,
>
> Submission 5758 Authors

---

### Author Response · Authors · 2025-11-18
**General Response to Reviewers**

We sincerely thank all reviewers for their valuable and constructive feedback, as well as for recognizing the novelty, completeness, clarity, and presentation of our work. We greatly appreciate their comments highlighting our contributions and the insightful suggestions that have helped us strengthen the rigor and readability of the paper.

Following the feedback from the reviewers, we have made the following major revisions and additions:

- Additional Experiments
  - We conducted new experiments on Qwen2.5-7B-Instruct to demonstrate the general effectiveness and scalability of CoMAS on larger models.
  - We included SRLM as an additional baseline to better contextualize CoMAS against direct self-evolution approaches.
  - We re-evaluated all methods under the Vanilla setup, reporting the mean and standard deviation over 5 random seeds to demonstrate the statistical significance of performance improvements.
  - We added an analysis of training cost to more comprehensively discuss efficiency aspects.
  - We analyzed the consistency between the evaluation accuracy and the precision and recall of the generated rewards to further validate the effectiveness of our adversarial reward design in mitigating reward hacking.

- Presentation Improvements
  - We substantially refined the Related Work section by grouping prior studies according to their methodology and learning paradigm, thereby clarifying how CoMAS differs from existing approaches and emphasizing its novelty.
  - We clarified the text and caption in Figure 2 to eliminate ambiguity regarding the use of heterogeneous LLMs and to better illustrate the progression of the three interaction stages.
  - We increased the font size and improved the readability of Figures 3, 4 and 5.
  - We explicitly stated the consistent use of REINFORCE++ across all methods following MARTI.
  - We added examples of failed discussions arising from ablations that remove critical components, enhancing completeness and transparency.

We hope these substantial revisions adequately address the concerns from the reviewers and further strengthen the quality of our work. We would be deeply grateful if you could consider these improvements in your final evaluation. Please feel free to let us know if any additional clarification or analysis would be helpful.

---

### Comment · Area_Chair_LpGS · 2025-11-27
**Please review the authors' responses and provide feedback ASAP**

Dear Reviewers,

Thank you for your essential contributions to the review process. The authors have submitted their responses to your initial reviews.

I kindly ask you to carefully review the authors' responses for this submission. Your timely assessment of how the authors have addressed your original concerns is a critical step in reaching a final decision.

Please provide your feedback and any necessary updates to your reviews as soon as possible to ensure we can meet our tight schedule for the discussion phase.

Your prompt attention to this matter is highly appreciated.

Regards,

-AC

---

### Meta-Review · Area_Chair_k7rR · 2026-01-04

**Summary:**

I will list the most important comments that the reviewers noted during the review process:
1) The performance gap is only significant in a minority of the tasks
2) The reward formulation doen’t solve the reward hacking problem
3) Obtaining rewards through multi-agent discussions is not a new idea
4) The Related Work section lacks detailed and structured discussion of prior work
5) Most comparisons are conducted under the paper’s unified pipeline
6) Most experiments are based on homogeneous LLMs
7) The choice of Qwen2.5-3B-Instruct as the base model imposes a fundamental constraint on CoMAS’s evolutionary potential
8) The paper lacks critical analysis of how interaction complexity impacts evolution efficiency
9) Missing comparisons with other recent self-evolution methods mentioned in related work
10) Experiments only go up to 4 agents; unclear how the framework scales to 10+ agents

**Reviewer Concerns:**

The authors did a lot of work during the rebuttal phase and addressed most of the reviewers' comments:
1) The performance gap: authors repeated the evaluation under the Vanilla setup using 5 random seeds, reporting mean and standard deviation across runs.
2) The reward formulation: authors align signals with rule-based verifier results, treat the verifiers as ground truth, and compute precision and recall of generated rewards.
3) The Related Work section: authors have reorganized it to group prior studies by methodology and learning paradigm.
5) Unified pipeline: the comparisons representative within standard community practices.
6) Homogeneous LLMs: second ablation study already examines the heterogeneous setting.
7) Larger models: authors onducted additional experiments using Qwen2.5-7B-Instruct under the same settings.
8) Critical analysis: authors provided additional analysis.
9) Missing comparisons: authors trained SRLM on dataset under identical settings.

Despite this work, some of the comments remain valid:
1) Obtaining rewards through multi-agent discussions is not a new idea: the authors have positioned their work well enough and the novelty is clearly marked.
2) Missing comparisons: there are still relevant methods that could be compared.
3) Scalability with More Agents: the authors have reserved this remark for future work.

**Reviewer Scores:**

1) Reviewr bqfG (score 6) could raise his score.
2) Reviewer F6Fu (score 4) could raise his score.
3) Reviewer CxUs (score 4) could raise his score.
4) Reviewer qMBz (score 6) would most likely have left his initial score.

---

### Decision · Program_Chairs · 2026-01-26

Accept (Poster)